# The Influence of Calcareous Fly Ash on the Effectiveness of Plasticizers and Superplasticizers

**DOI:** 10.3390/ma13102245

**Published:** 2020-05-13

**Authors:** Jacek Gołaszewski, Tomasz Ponikiewski, Aleksandra Kostrzanowska-Siedlarz, Patrycja Miera

**Affiliations:** Department of Building Processes and Building Physics, Faculty of Civil Engineering, Silesian University of Technology, 44-240 Gliwice, Poland; Jacek.Golaszewski@polsl.pl (J.G.); Aleksandra.Kostrzanowska-Siedlarz@polsl.pl (A.K.-S.); Patrycja.Miera@polsl.pl (P.M.)

**Keywords:** calcareous fly ash, plasticizer, superplasticizer, rheological properties, fly ash processing methods, cement mortars, workability

## Abstract

Due to the rational shaping of the environment and the management of environmental resources in accordance with the principle of sustainable development, calcareous fly ash (CFA)—high-calcium as a by-product of lignite combustion—is a valuable addition to concrete. This additive, however, due to its high-water demand lowers the workability of the concrete mix, which is a problem, especially in the first 90 min after mixing the components of the mix. In order to meet this challenge, plasticizers (P) and superplasticizers (SP) for concrete are used with various effects which are designed to reduce the yield value and plastic viscosity. To check the technical efficiency of admixtures P and SP with different chemical bases, the main objective of this research was to investigate the influence of raw and ground CFA on the rheological properties and other side effects of admixtures, such as the amount of air in the mixture and the amount of heat of hydration. The use of P, particularly SP, effectively improves the workability of the mortar containing CFA, especially ground CFA. With these admixtures, it is possible to obtain mortars containing ground CFA with similar rheological properties to mortars without its addition. To obtain a specific workability of mortar with CFA, it is usually necessary to introduce a higher dose of P or SP than used for mortars without CFA. The presence of raw CFA does not alter the effectiveness of P and strongly reduces the effectiveness of SP. The reduced effectiveness of SP manifests primarily as a high workability lost. The presence of ground CFA does not change the effectiveness of P (or is higher). The effectiveness of the superplasticizer SNF (with a chemical base of naphthalene sulfonate) and PE (with a chemical base of polycarboxylate ether) is slightly lower or does not change. The effectiveness of the superplasticizer SMF (with a chemical base of melamine sulfonates) is significantly lower. We found that the presence of ash affects the efficiency of P and SP, while processing via the grinding of ash makes the effect negligible. These results are novel in both their cognitive and practical aspects.

## 1. Introduction

Waste management for the coal fired power plants is gaining key importance in connection with threats to the environment and health. Such power plants in Poland alone produce millions of tons of fly ash per year, whose properties depend on the type of coal (lignite or hard coal) and the method of combustion. The reuse of this fly ash in the composition of concrete relates to sustainable development by reducing the amount of cement and thereby reducing cement production, which is associated with a reduction in the amount of CO_2_ released into the atmosphere. In addition to this ecological aspect, there is also an economic one, as fly ash can provide measurable benefits to investors by replacing cement and clinker with waste materials. Fly ash from the combustion of hard coal is characterized by its pozzolanic properties, and due to its beneficial effect on the properties of concrete, it is a valued and widely used addition to concrete. When lignite is burned, calcareous fly-ash (CFA) is produced—i.e., fly ash that contains large amounts of calcium compounds. However, due to its composition and physical properties, not every CFA is suitable for concrete [1]. The studies carried out to date [2] show that, among the CFA available in Poland, only the CFA from the Bełchatów power plant has properties that allow it to be used as an additive to cement and concrete. This CFA has high pozzolanic and hydraulic activity and meets the requirements of the EN 197-1 standard [3] for the main constituents of common cement. After the grinding process, it can also be used as an active mineral additive to concrete [1,3,4,5]. As established in several studies [6,7,8,9,10,11,12], the use of this CFA in up to 30% of cement as an additive to concrete or as the main component of cement, in general, does not negatively influence the strength and durability of the concrete. Notably, it is necessary to use CFA processed by grinding, not in a raw state. Moreover, the authors in [13] investigated the effects of nanoclay additions on the fresh properties, mechanical performance, and microstructure properties of high volume fly ash mixes designed for 3D printing. The results in [13] showed that the addition of high volume fly ash improved the thixotropic properties of the mixtures, thus increasing its suitability for concrete printing applications.

Unfortunately, the use of CFA as a concrete additive is significantly hindered by problems related to fresh concrete’s workability. In a raw state, CFA is characterized by very high water demands, much higher than those of cement [5,14]. These demands can be reduced by processing, preferably by grinding [2,14]. Even then, the water demands remain higher than those of cement [2,6]. The high water demands of CFA make it difficult to obtain fresh concrete with the required stable workability in the long term, whether it is used as an additive to cement or as an additive to concrete [6,12,13,14,15,16,17,18]. The processing of CFA solves this problem to some extent, but it should be noted that the use of ground CFA undoubtedly has a beneficial effect on the rheological properties of mortars and their variability [12,14,16,17,18].

Therefore, to obtain the required workability of CFA containing concrete, it is necessary to use plasticizers (P) or superplasticizers (SP). Indeed, the possibility of using CFA is conditional on the use of these admixtures [12,14,18]. Therefore, the effectiveness of these admixtures in the presence of CFA is particularly important. However, the experimental data on this topic has been limited. In general, to obtain a specific workability of fresh concrete with CFA, it is necessary to use more P or SP than for of the corresponding compositions without CFA [16,17,18,19,20,21,22]. This is likely due primarily to the higher water demand of fresh concrete with CFA. Consequently, there is a smaller amount of free water in the mixture [23]. The potentially lower efficiency of P and SP in the presence of CFA is indicated by the faster loss of workability of mixes with CFA [6], but such effects do not occur in every case [16,17,22]. To date, there has been no in-depth study on how different types of P and SP work with CFA with different properties, both in terms of the primary effect—the rheological properties of the mortars—and the secondary effects—setting time, air entrainment, or hydration heat. Generally, this indicates that present knowledge of the impact of CFA on the effectiveness of P and SP is very limited and not systematic; thus, further studies are needed. 

The main objective of this research was to investigate the influence of raw and ground CFA on the effectiveness of P and SP activity. The basic effect of P and SP on rheological properties was studied using rheometric techniques. The secondary effects of the admixture’s effects, such as setting time, heat of hydration, and air content, were also studied.

## 2. Effectiveness of Plasticizers and Superplasticizer Action

The effectiveness (efficiency) of concrete admixtures is a criterion based on the characteristics of the quality of its effects in its given function and its associated primary effect [24]. The primary effect is defined here as an effect of the admixture corresponding to its function as a direct consequence of the physical mechanism of its action. Typically, the assessment of the effectiveness of an admixture and its applications should take into account secondary effects because of the possible adverse impact of the admixture on the important properties of the concrete and (or) the hardened concrete. The types and primary and secondary effects of P and SP are summarized in Table 1.

The effectiveness of P and SP should be considered from technical, technological, and economic perspectives. Technical effectiveness determines the changes in the rheological properties of the fresh concrete in terms of the minimum dosage of admixture needed for its effects to take place in the intended time needed for transporting and arranging the mix at the installation site; the conventionally adopted time is 90 min. Economic effectiveness refers to the cost of obtaining certain changes in the rheological properties using the above additives. Technological effectiveness is the ease and safety of using the admixture and the sensitivity of its effects to changes in environmental conditions. This article focuses on the technical and rheological aspects of the effectiveness of P and SP in the presence of CFA. In practice, the choice of admixture also depends on economic and technological factors. The fulfilment of these factors will achieve the desired effect of the admixture at the lowest cost and in a safe manner. 

The aim of using P and SP is to adequately modify the rheological properties of the fresh concrete according to the technology used and the conditions for the implementation process of concreting. The basis for evaluating the effectiveness of these additives is measuring their impact on the changes of their rheological properties and workability. Therefore, the effectiveness tests of these admixtures focus primarily on the identification effect on the rheology of fresh concrete under certain technological conditions and the possibility of side effects of the admixtures, such as changes in the aeration of the mix or changes in the heat release curve during cement hydration. For this purpose, it is necessary to adopt a rheological model of the fresh concrete, and then measure the changes in its rheological parameters alongside the air content in the mix and changes in the nature of heat release during the hydration process, as a result of the addition of an admixture in terms of the variable factors and type of the concrete components.

Physically, mortar and concrete are similar. Both are a mixture of cement, water, aggregate, admixtures, and additives. Numerous studies show that the tests carried out on mortars can also be used to predict the rheological properties of fresh concrete. Simple mathematical relationships between the rheological properties of fresh mortars and fresh concrete mixes are presented in past studies [25,26,27,28,29,30,31,32,33]. Thus, it is commonly accepted that mortars can be used to test the effectiveness of P and SP. Thanks to this, the cost of research can be significantly reduced, and its scope can be increased. Therefore, studies on the effects of CFA on the performance of plasticizers and superplasticizers were also performed on mortars.

## 3. Experimental Section

### 3.1. Variables and Research Plan

The research plan is presented in Table 2. The research was conducted for three batches of CFA (raw fly ash; batches: A, B, C, and ground fly ash; batches: AG, BG, and CG), sampled in a time range of a half year from the intermediate reservoirs of the Bełchatów Power Plant. We used both raw and ground CFA, which were added as a substitute for 20% of the cement mass. The effectiveness of different admixtures was assessed by testing changes in the rheological properties of the mortars and the testing side effects of the admixture, including the heat of hydration and air content, with and without CFA. We selected two P and four SP that are typically used and represent the main types of this admixture. Admixtures were also selected based on their different chemical bases that were representative of the given admixture group: For P: lignosulfonates, iminodietanol, bis ethanol, phosphate (V) tri butyl acetate, formaldehyde, methanol, and (Z)-octadec-9-enyloamine; for SP: polycarboxylate ether, melamine sulfonates, and naphthalene sulfonate.

The maximum amount of P and SP corresponded to the maximum amount recommended by the producer of the admixture. The maximum content of admixture also did not exceed the saturation point, which was verified in preliminary studies.

This study was conducted on mortars, but due to the similarity of the rheology of mortars and concrete mixes, it can also be used to design the workability of a concrete mix.

### 3.2. Materials and the Composition

The composition and selected physical properties of the raw and ground CFA used in this research are compiled in Table 3. Blaine specific surface was tested according to [34].

Ground CFA was created by subjecting raw CFA to a grinding process in a laboratory ball mill. The residue on the 45 μm sieve was taken as the measure of grinding. Due to its coarse granulation and value of fineness (minimum, 36%; average, 50%), the tested CFA did not meet the basic requirements set by the ASTM C618 standard [35] (retention on a 45 μm sieve at a maximum of 34.0%) and PN-EN 450-1 standard [36] (retention on a 45 μm sieve at a maximum of 40%). The other requirements for the CFA composition were, however, met. Fluctuations in the chemical composition and properties of the ash are significant, especially the amount of CaO, SO_3,_ and Na_2_O. However, it should be noted that CFA is characterised by a relatively low changeability in the amount of SiO_2_ and Al_2_O_3_ and a low loss on ignition. The X-ray diffraction (XRD) pattern of CFA is presented in Figure 1. The differential thermal analysis (DTA) pattern of CFA is presented in Figure 2, Figure 3 and Figure 4. The cumulative distribution of ash grain size is presented in Figure 5. This ash contains above 25% reactive silica and above 10% reactive calcium oxide, which shapes its pozzolano–hydraulic properties. The results of the supplementary tests in terms of phase composition and granulation confirm the above-mentioned observations on the usefulness of calcareous fly ash as a pozzolan–hydraulic component of cement for batches of materials with different phase compositions (see the diffractograms and thermograms in Figure 1, Figure 2, Figure 3 and Figure 4) and variable particle sizes within the fluctuations shown during intensive monitoring, as shown in Figure 5. Observations of calcareous fly ash using scanning electron microscopy showed the presence of grains with a spherical shape and a smooth surface, as well as irregularly shaped porous grains, as displayed in Figure 6, Figure 7 and Figure 8.

After processing by grinding, the requirement of fineness under 34% is always met, and the Blaine specific surface is 3500–3700 cm^2^/g. The water demand of the tested CFA is high. Replacing 20% of the cement with CFA causes the water demand to increase from 8% to 12% (on average, 10%) (the test procedure according to PN EN 450-1 [36]). Processing of the CFA by grinding causes the water demand to decrease. Replacing 20% of the cement with ground CFA causes the water demand to increase from 2% to 6% (on average, 4%). 

The properties used for P and SP are presented in Table 4. The properties of the CEM I 42.5 cement used in this research are presented in Table 5. The mortar proportions are shown in Table 6. In order to eliminate the influence of the type and grading of sand on the rheological properties of the mortars, normal sand (2 mm maximum with a bulk density of 2.65 g/cm^3^, according to PN-EN 196-1 [37]) was used. The grading curve of the normal sand is presented in Figure 9. The proportions of the mortar mixture were based on standard mortar proportioning according to PN-EN 196-1 [37] but with the w/b ratio changed to 0.45 or 0.55.

### 3.3. Testing Effectiveness of the Plasticizer and Superplasticizer Action

#### 3.3.1. Rheological Properties

The rheological behaviour of mortar and concrete is commonly described by the Bingham model using the parameters of yield value and plastic viscosity.

The yield value determines the shear stress necessary for initiating flow. When the shear stress is higher than the yield value, the mixture starts to flow at a speed inversely proportional to the plastic viscosity. The yield value controls the workability of ordinary fresh concrete, while the role of plastic viscosity is secondary. In the case of self-compacting concrete characterized by a low yield value, the plastic viscosity determines the flowability, stability, and ability to self-deaerate. Problems with the rheology of mortars and concretes are discussed in detail in [28,38,39].

The mortars for testing the rheological properties were prepared according to PN-EN 196-1 [37], and the mixer and mixing procedures were compliant with PN-EN 196-1 [34]. CFA was added together with cement, and the admixtures were added to water (PE) and delayed for 30 s (P, SNF, and SMF). After the end of mixing, the mortar samples were transferred to a Viskomat NT rotational rheometer. The rheological parameters g (Nm) and h (Nm s), corresponding to yield value and plastic, were then determined. The values of g and h can be presented in physical units, but the measurement constants of the rheometer have to be defined. According to [29], in an apparatus like the one used in this work, τ_o_ = 7.9 g and η_pl_ = 0.78 h. However, since the rheometer constants were not verified, the results are presented as g and h. The mean relative errors of determination of the rheological parameters g and h of the mortars containing CFA were, respectively, 4.4% and 4.5%, which are identical to other studies. This proves that the Bingham model is acceptable for describing the rheological properties of mortars with CFA and P or SP. The general basis and rules for rheological measurements are detailed in [38,39]. The tested mortars were prepared and stored between measurements under conditions that allowed its temperature to remain at 20 °C. During the measurements, the required temperature was maintained with an automatic thermostatic controller.

#### 3.3.2. Air Content

The air content in the mortar was determined by PN-EN 1015-7 [40].

#### 3.3.3. Heat of Hydration

The heat of hydration for the cement–CFA–admixture systems was determined using an isothermal microcalorimeter (TamAir). This apparatus measures the amount of heat (in J/g) that is emitted under isothermal conditions during binder hydration (CFA and CEM I) from the moment of its contact with water and admixture in relation to an inert referential sample with an analogous heat capacity. The water–binder ratio (w/b) of the tested cement paste was 0.45 (P, SMF) or 0.55 (SNF, PE). This measurement was conducted on a binder sample weighting 5 g, mixed with 2.25 g or 2.75 g of water. During the measurement, the temperature of the cement paste was 20 °C. The measurement of the heat of hydration lasted 12 h.

## 4. Results and Discussion

The influence of P and SP on the rheological properties of CFA mortars is shown in Figure 10, Figure 11, Figure 12 and Figure 13, and their influence on the air content and heat of hydration is shown in Table 7 and Table 8, respectively.

The PL and SP used in this study liquefied the cement mortars containing CEM I without the addition of CFA. Studies have shown that the workability and stability of mortar are retained for a period of 90 min. Thus, the admixtures used are compatible with the cement used in the study.

Adding raw CFA to mortars as a cement replacement causes a significant increase in the yield value g and plastic viscosity h, depending on the type of CFA. The range of changes in the yield value g of mortars increases over time, but the presence of CFA insignificantly affects changes in the plastic viscosity h over time. The nature of the influence of ground CFA on the rheological parameters of mortars is the same as that of raw CFA. It also worsens the workability of mortar, however, to a much lesser extent than raw CFA. The influence of CFA type and processing method is presented and discussed in [14,19].

In order to determine the significance of the influence of the compositional factors and their interactions on the rheological properties, an analysis of variance (ANOVA) was carried out using one-dimensional significance tests for the rheological parameters (g5, g90, h5, and h90) of the mortars with P and SP. The results are shown in Table 9 and Table 10, which present the ANOVA with parameterization, sigma-restrictions, and a decomposition of the effective hypotheses. The ANOVA statistical analysis showed that the largest statistical effects on rheological parameters were yield value and plastic viscosity, regardless of the time at which the measurement was made, and the type of batch. The rheological parameters of the mortars with P were also affected by the type of batches and type of P. However, the rheological parameters of the mortars with SP were affected by the dosage of SP.

In the presence of P1 and especially P2, the negative influence of raw CFA on the yield value g of mortars is clearly lower. After the addition of 0.5% P1 or P2, the yield value g of the mortars with ground CFA is usually lower than that of mortar without CFA. For P1, this effect disappears over time, while for P2, it remains strong after 90 min. P1 has an insignificant effect on the plastic viscosity h of the CFA mortar. P2 significantly reduces plastic viscosity h. The addition of plasticizers makes the changes in the plastic viscosity h of the CFA mortars less significant over time. The results obtained are consistent with the results of the tests on cements containing CFA in [16,17]. 

However, the increased SMF addition yield value g of the CFA mortars generally remains higher than that of the mortar without CFA until the maximum recommended dose of SMF is applied (only for mortars with ground CFA). The presence of SMF accelerates the increase in yield value g for mortars over time. This increase is greater for mortars containing CFA. In general, the impact of SMF on the plastic viscosity h of the tested mortars is insignificant from a workability point of view.

The test results agree with those in [24,41,42], in which it appears that admixtures based on melamine sulfonates demonstrate a possible decrease in the water content in concrete by up to 20%–30% compared to synthetic polymers, such as polycarboxylates and acrylic copolymers (PCEs), which have versatile chemical structures and can achieve up to 40% water reduction.

Obtaining consist CFA mortars to measure their rheological properties requires the addition of 1.8% SNF. When raw CFA is used, the yield values of these mortars range from 2.5 to 3.5 times higher than the yield values g of the reference mortars, but when ground CFA is used, the yield value g of mortars only ranges from 1.5 to 2 times higher. Increasing the amount of SNF to 3.6% causes the yield value g of the mortars with ground CFA to drop lower than that in the reference mortar (an average of 15%). Mortars with raw CFA are then characterized by an average yield value g higher than 75%. At a dose of up to 2.4%, the SNF range of changes over time for the yield value g of mortars with CFA is clearly higher than that of the reference mortar. When 3.6% SNF is used, the increase in the yield value g over time for the reference mortar and mortars with ground CFA is negligible. To a small extent, the amount of SNF in the mortars with and without CFA influences the plastic viscosity h. The range of changes in the plastic viscosity h over time for the mortar and mortars with ground CFA is low and shows no clear trend; the workability these changes can thus be considered negligible. For mortars with raw CFA, with 2.4% SNF, they show a large decline in their plastic viscosity h resulting from stiffening of the mixture [26]. 

Obtaining consistent CFA mortars to measure the rheological properties requires the addition of 1% PE1 or 0.5% PE2. When raw CFA is used, the yield value of these mortars ranges from 2.2 to 3.4 times higher than the yield value g of the reference mortars, but when the ground CFA is used, the yield value g of the mortars only ranges from 1.25 to 2 times higher. The increase in the yield value g over time for the raw CFA mortars is very high, and after 90 min, these mortars are too stiff to perform rheological measurements. The increase in the yield value g of ground CFA mortars is usually clearly higher than that of the reference mortar (from 1.5 up to 2 times), but in some cases, it can be similar (PE1 or AG). The plastic viscosity h of the mortars with both ground and raw CFA is similar to or slightly higher than that of the reference mortar, and the plastic viscosity h of the PE1 mortars is higher than that of the PE2 mortars. The plastic viscosity h of the P1 mortars generally does not change in 90 min, while that of the PE2 mortars decreases. 

Further increasing the amount of PE1 and PE2 reduces the yield value g and plastic viscosity h. This reduction is higher for mortars containing CFA. With the addition of 2% PE1 and 1% PE2, the rheological properties of the reference mortar and the mortars with ground CFA are similar (sometimes the yield value g of ground CFA is even lower), and the mortars do not show significant changes in their yield value over time. For the raw CFA mortars, the yield value g and its growth over time are reduced by increasing the SP addition but remain considerably higher than those of the reference mortar. Increasing the dose of PE1 insignificantly influences the plastic viscosity h of the mortars. This is due to the properties of raw CFA and the high water demands, which were confirmed in [16,22]. Increasing the dose of PE2 lowers the plastic viscosity h of the mortars with ground CFA and increases the plastic viscosity h of the mortars with raw CFA. The plastic viscosity h of the mortars with raw CFA with PE2 at a dose of 0.75% significantly decreases, and with a dose of 1.0% PE2, it insignificantly increases.

The effects of P and SP action are affected by the type of CFA. On the basis of the conducted studies, it is not possible to identify clear trends. However, the use of P and SP reduces the influence of the type of CFA (particularly when the CFA is ground ) on the rheological properties of mortars (particularly when large amounts of P and SP), but the influence of CFA type may still be noticeable even if the maximum recommended dose is used.

The introduction of P1, SMF, and PE1 does not aerate the mortars, while the use of P2, SNF, and PE2 does aerate the mortars, both with and without CFA, as shown in Table 7. This effect may be partially responsible for the relatively smaller plastic viscosity of the mortars with the addition of P2, SNF, and PE2. 

The introduction of P2, SMF, SNF, and PE1 reduces the cement hydration heat emitted after 2 h by 60–80%, as shown in Table 8. These results are consistent with those of other studies in this field [43]. In the presence of CFA, the reduction in the amount of heat released by adding these admixtures is smaller and ranges from 10% to 45%, depending on the nature and processing of the CFA (without showing clear trends). This indicates the retarding effect of admixtures, which is lower in the presence of CFA. The reasons for this can also be seen in the mechanism of the increased absorption of P and SP described above by large, irregular CFA grains. This phenomenon causes a smaller amount of P and SP to act in the cement paste, thereby exerting a smaller effect on the hydration process. It should be noted that a reduction in the heat generated after 2 h and 12 h by PE1 is higher than that for SMF and SNF. This indicates the strong retarding effect of PE1. 

## 5. Evaluation of the Effectiveness of Plasticizers and Superplasticizers in the Presence of CFA

Evaluation of the effectiveness of P and SP in the presence of CFA was focused on the changes in the yield value g of mortars. Thus, the initial yield value g and its changes over time were taken into account. Plastic viscosity h, as indicated earlier, is normally of secondary importance to the mixture’s workability. Additionally, as shown in this research, the range of plastic viscosity h changes in mortars due to the addition of P or SP with or without CFA is, in most cases, insignificant.

The obtained results for PL and SP activity do not indicate that the presence of CFA significantly affects their mechanisms of action described in [24,44]. The introduction of CFA as a cement replacement, due to its increased water demands, reduces the amount of free water in the mixture. Accordingly, mortars with CFA are characterized by a much higher yield value g and a faster increase in the yield value g over time than in mortar without CFA. Thus, to obtain a certain yield value g of mortars with CFA, it is necessary to use a higher addition of P or SP than for similar mortars without CFA. The amounts of P1 and P2 and SNF, SMF, and PE necessary to obtain a mortar yield g equal to 20 Nm are shown in Figure 14a and Figure 15a. These relationships demonstrate the beneficial effect of using ground CFA. Obtaining the specific yield value g of ground CFA mortars requires a significantly lesser amount of admixture than that of raw CFA mortars. Importantly, it also shows that immediately after mixing, in the presence of ground CFA, plasticizers P1 and P2 are more effective, while the superplasticizers SNF, PE1, and PE2 and SMF are significantly less effective than in mortars without CFA. Only the effectiveness of P1 and SMF depend on the type of CFA; the effectiveness of the other types of admixtures, to a lesser extent, depends on the type of CFA, especially when the CFA is ground. The increase in the yield value g of mortars with an initial yield of 20 Nm is shown in Figure 14b and Figure 15b. This increase is generally much higher for mortars with CFA, especially when raw CFA and SMF and PE2 are used. Only for P2 and PE1 is the increase in the yield value g over time for mortars with ground CFA less than or similar to that for mortars without CFA. This means that the effectiveness of P and SP with respect to time of action is generally reduced in the presence CFA. Analyzing the available literature [1,24,42,44,45] shows that the morphology of CFA grains affects the lower efficiency of P and SP. Raw CFA is characterized by large, porous grains with a large developed surface, which also contain large porous residues of unburned coal. This is the reason for the increased absorption of P and SP on CFA grains. This phenomenon significantly reduces the amount of admixtures that can work effectively in a cement mix. During processing by grinding, large grains are destroyed, which both reduces the CFA’s water demand [14] and contributes to an increase in the amount of active P or SP. The CFA processed by grinding increases the effects of the admixtures in comparison with the operations in cement mixes modified by raw CFA.

For mortars with CFA, the effectiveness of P and SP in the presence of CFA was also analyzed according to the changes in the initial yield value g5 and the increase in the yield value g over time of up to 90 min (g90–g5) caused by the addition of these admixtures compared to the analogous changes of (i) the reference mortar (without CFA) and (ii) the CFA mortar without an admixture. The relative influence of CFA type and processing on the effectiveness of P and SP is shown in Figure 16 and Figure 17.

The presence CFA favourably impacts the initial effectiveness of P1 and P2. The range of the reduction of the yield value g caused by the addition of P is higher in the mortars with CFA, particularly in mortars with ground CFA. The yield value g of the ground CFA mortars with a P addition of 0.25% is always lower than that of the mortars without CFA. The presence of CFA negatively affects the effectiveness of P1 with respect to workability changes over time. The relative increase in the yield value g over time for all mortars with CFA and P1, but especially those with raw CFA, is significantly higher than that for analogical mortar without CFA. At the same time, the presence of CFA favourably impacts the effectiveness of P2. The relative increase in the yield value g over time for mortars with P2 and with unprocessed and (particularly) ground CFA is lower than that of the mortar without CFA. Thus, using processing with CFA increases the effectiveness of P. 

The effectiveness of SMF in the presence of CFA is clearly lower. The relative reduction of the initial yield value g of the CFA mortars is lower than of the mortar without CFA, even when the ground CFA is used. It should be noted, however, that despite lower effectiveness in the presence of CFA, the effects of SMF action remain higher than those of P1 and P2. The effects that adding SMF have quickly disappear over time (faster than for P1 and P2), which is typical for this type of admixture [24]. With the addition of 1.15%, the SMF increase in the yield value g of the mortars with ground CFA is clearly higher than that in the reference mortar and even higher than that in mortars without the addition of SMF. This means that the effectiveness of SMF with respect to time is vulnerable to CFA, especially raw CFA.

The initial effectiveness of the SNF in the presence of ground CFA does not reduce significantly but, at the same time, is reduced in the presence of raw CFA. Thus, at an SNF dose close to maximum, the mortar with ground CFA has a lower yield value g than the mortar without CFA. For workability loss, the effectiveness of SNF in the presence of CFA (both raw and ground) is reduced. Only at a dose of 3.6% SNF (the maximum recommended dose) was it possible to obtain ground CFA mortars with the range of changes in yield value g over time analogous to those of the mortar without CFA. In conclusion, the presence of CFA reduces the effectiveness of SNF. This reduction is lower when ground CFA is used.

The initial effectiveness of PE1 in the presence of ground CFA is higher but decreased in the presence of raw CFA. At 2% and higher dosages of PE1, the mortars with ground CFA achieve a similar yield to the mortars without CFA. In terms of workability loss, the effectiveness of PE1 in the presence CFA is lower. However, it should be noted that at high PE1 dosages, the workability loss of the ground CFA mortars and the reference mortar is negligible. On the other hand, the mortars with raw CFA show a considerable loss of workability even when the maximum recommended dose of PE1 is used. Thus, the presence of raw CFA reduces the effectiveness of PE1, but the presence of ground CFA affects it much less significantly.

The effectiveness of PE2 is generally lower than that of PE1. The presence of CFA, especially raw CFA, reduces the effectiveness of PE2. At a dose of 0.50%, PE2 was able to fluidize the raw CFA mortar only to a small extent. The mortars with raw and ground CFA present a rapid workability loss—much faster than that of the reference mortar. Increasing the dose of PE2 slightly reduces the yield value g of mortars with raw CFA, but even at its maximum recommended dose, such mortars show a rapid workability loss. An increased dose of PE2 strongly influences the reduction of the yield value g of mortars with ground CFA. At the maximum dosage, the yield values of these mortars are smaller than those of the reference mortar. The mortar with ground CFA still shows a rapid loss of workability. Therefore, in general, the presence of CFA negatively impacts the effectiveness of PE2, but to a lesser degree when ground CFA is used. 

The type of CFA affects the efficiency of all tested SP. However, based on the current research, it is not possible to identify clear trends (SP usually works clearly better in the presence of CFA-type A and worse with CFA type C, but this effect cannot be clearly linked to the specific properties of the CFA). With an increased amount of SP, the influence of the type of CFA on the rheological parameters of the mortar is reduced. However, for raw CFA, even at the maximum recommended dosage, this influence remains evident.

## 6. Conclusions

We confirmed that the use of raw CFA has a very negative impact on the workability of mortars. This effect is much less if ground CFA is used. The practical application of this ash without the simultaneous use of plasticizers or superplasticizers can be difficult in many cases.

The use of an admixture, particularly SP, allows one to effectively control the workability of mortar containing CFA, especially ground CFA. With these admixtures, it is possible to obtain mortars containing ground CFA with similar rheological properties to mortars without this addition. To obtain a specific workability of mortar with CFA, it is usually necessary to introduce a higher dose of a P or SP than found in mortar without CFA.

The presence of CFA also influences the effectiveness of the P and SP. This effect depends mostly on the rheological admixture type and CFA processing. Table 11 presents the general results of the impact of raw and ground CFA additions on the technical effectiveness of P and SP. With a value of (−1) for obtaining the specified effect of mix workability, a higher admixture dosage is necessary, and with a value of (−2), a higher admixture dosage is necessary (or the specified mix’s workability may be impossible).

The properties of CFA have an impact on the effectiveness of P and SP; this effectiveness is clearly lower when ground CFA is used. In the presence of CFA, the secondary effects of using P or SP for air entrainment or setting the time are similar. However, the effects of these admixtures on the heat of hydration are lower in the presence of CFA.

The obtained results may be used as an indicator for admixture selection or for the workability design of fresh mortars and concretes containing CFA. The use of P or SP allow one to effectively use CFA in concrete technology as a concrete or cement additive and thereby obtain significant environmental benefits. However, the selection of specific P or SP should always be verified experimentally while taking into account the CFA and cement’s properties, as well as the specific demands of the mixture’s workability and the secondary effects of the admixture.

## Figures and Tables

**Figure 1 materials-13-02245-f001:**
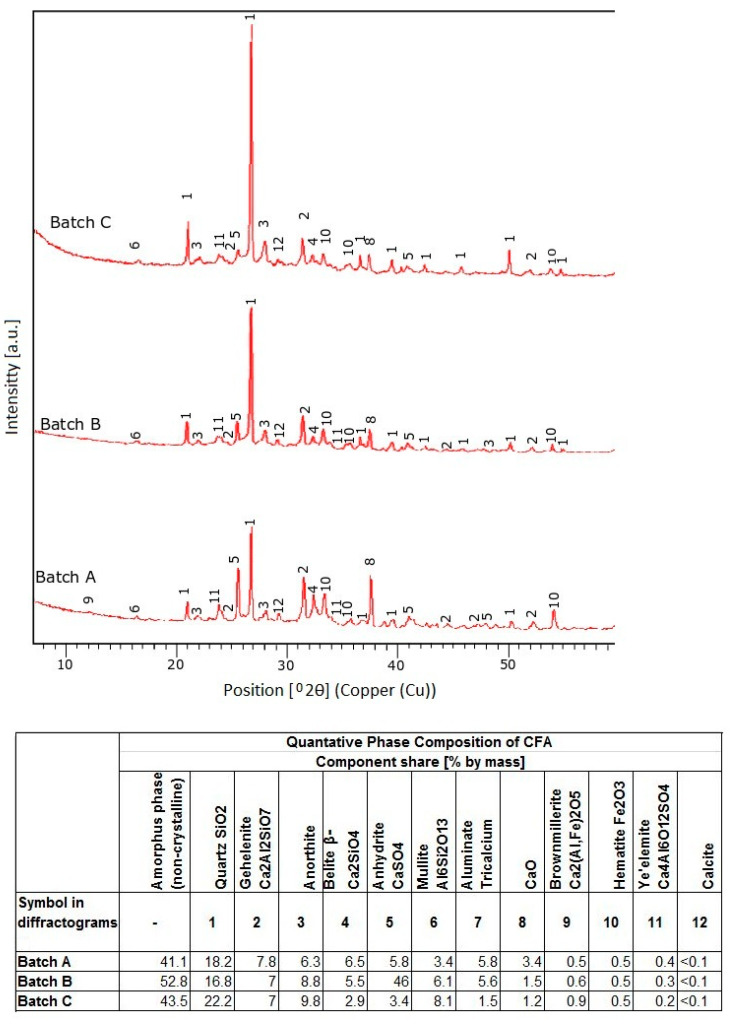
The X-ray diffraction (XRD) pattern of CFA batch A, B, C.

**Figure 2 materials-13-02245-f002:**
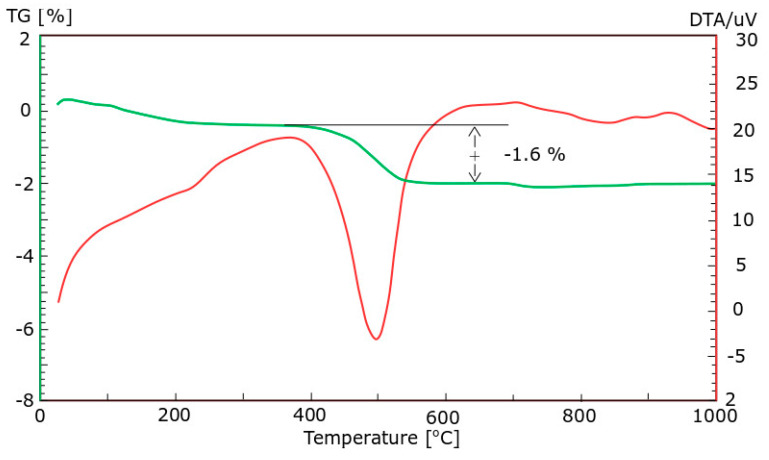
The differential thermal analysis (DTA) pattern of CFA batch A.

**Figure 3 materials-13-02245-f003:**
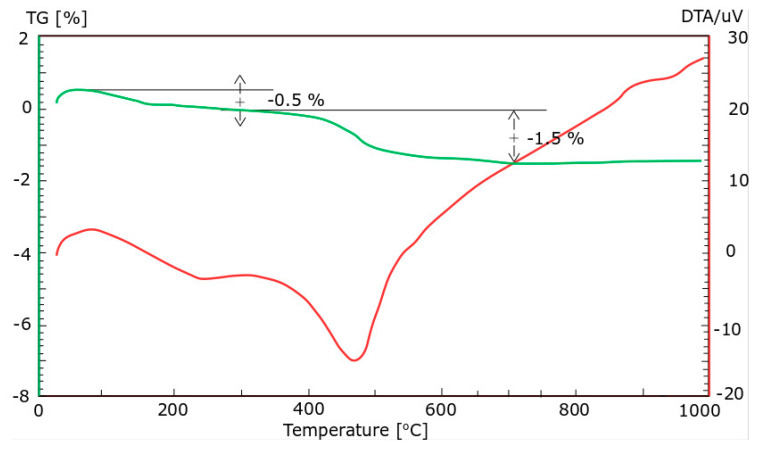
The differential thermal analysis (DTA) pattern of CFA batch B.

**Figure 4 materials-13-02245-f004:**
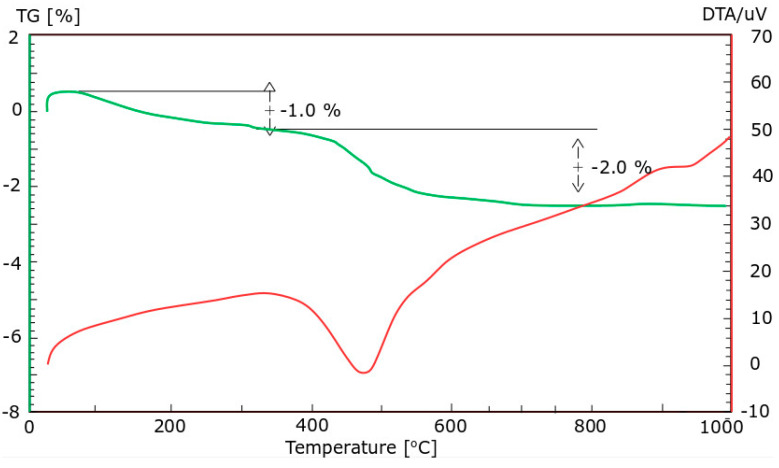
The differential thermal analysis (DTA) pattern of CFA batch C.

**Figure 5 materials-13-02245-f005:**
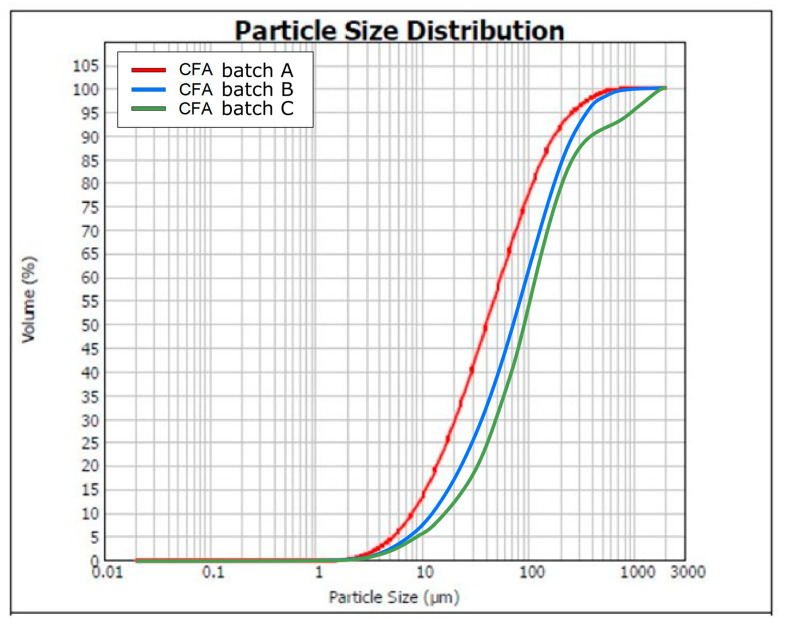
Cumulative distribution of ash grain size of CFA batch A, B, C.

**Figure 6 materials-13-02245-f006:**
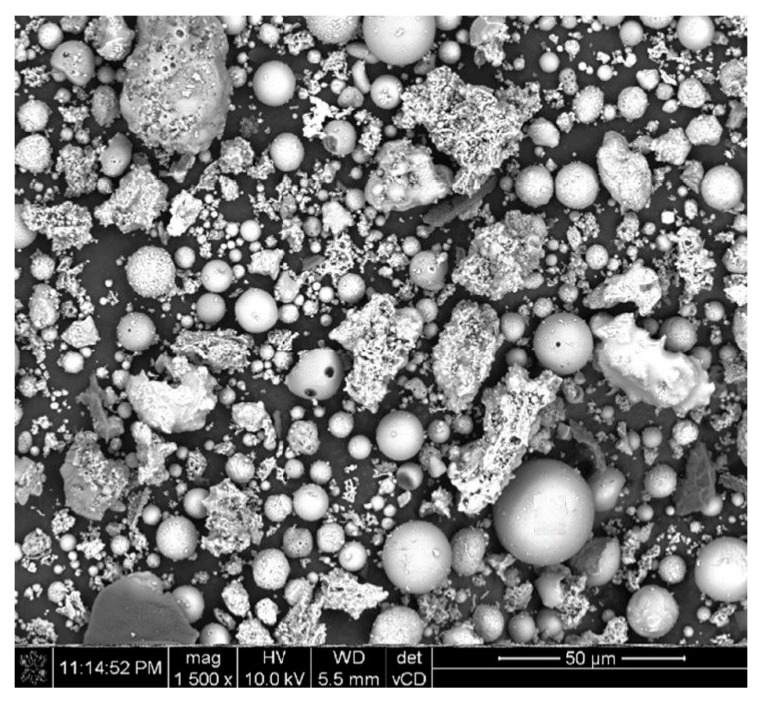
Morphology of calcareous fly ash grains batch A (magnification of 1500 times).

**Figure 7 materials-13-02245-f007:**
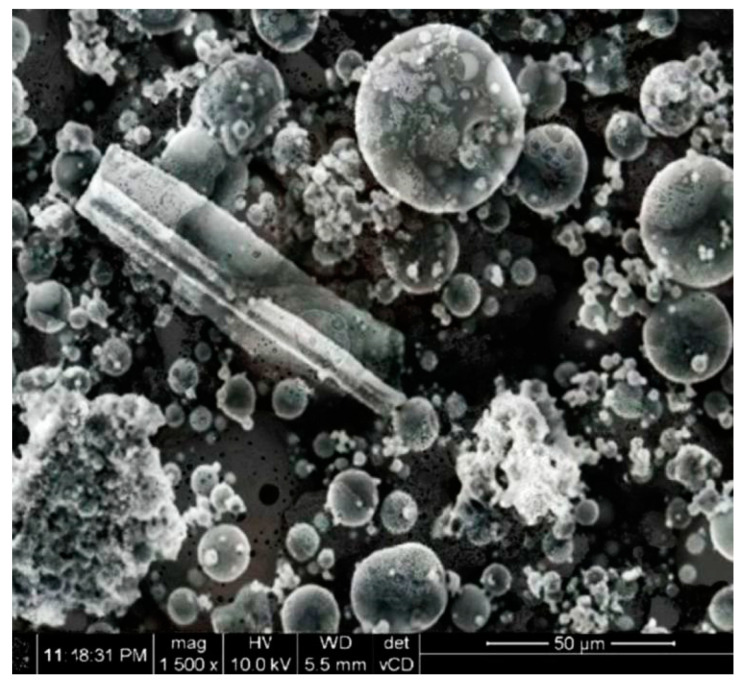
Morphology of calcareous fly ash grains batch B (magnification of 1500 times).

**Figure 8 materials-13-02245-f008:**
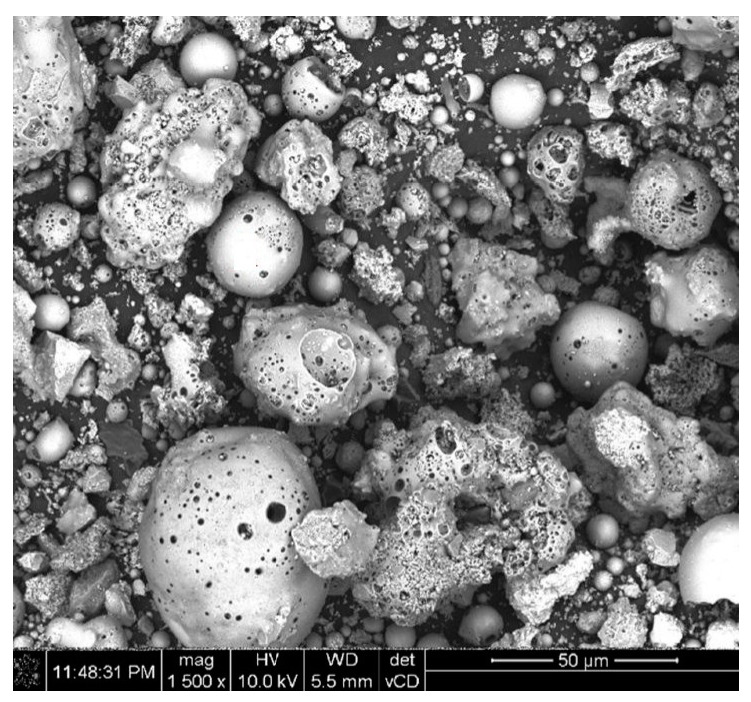
Morphology of calcareous fly ash grains batch C (magnification of 1500 times).

**Figure 9 materials-13-02245-f009:**
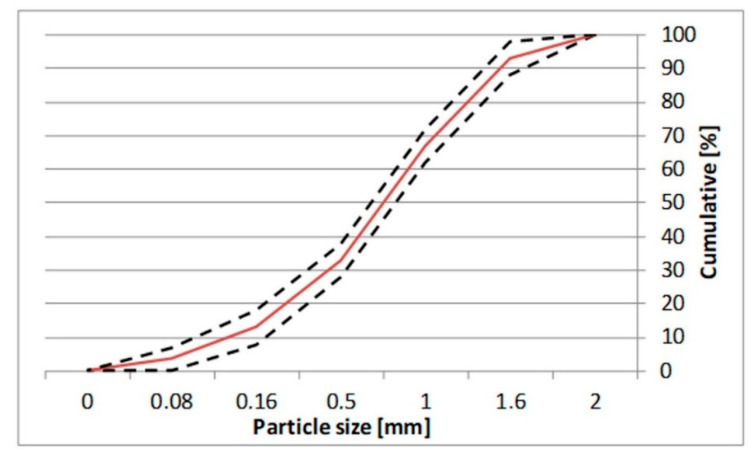
The grading curve of normal sand [34].

**Figure 10 materials-13-02245-f010:**
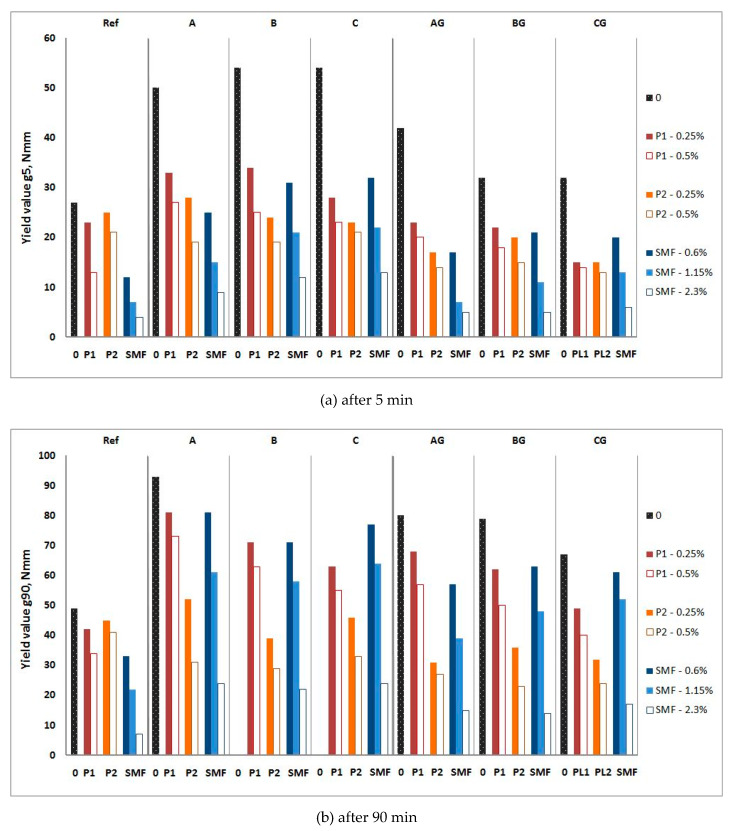
Influence of P1 and P2 and SMF on yield value g of mortars with raw and ground CFA. (**a**) after 5 min; (**b**) after 90 min.

**Figure 11 materials-13-02245-f011:**
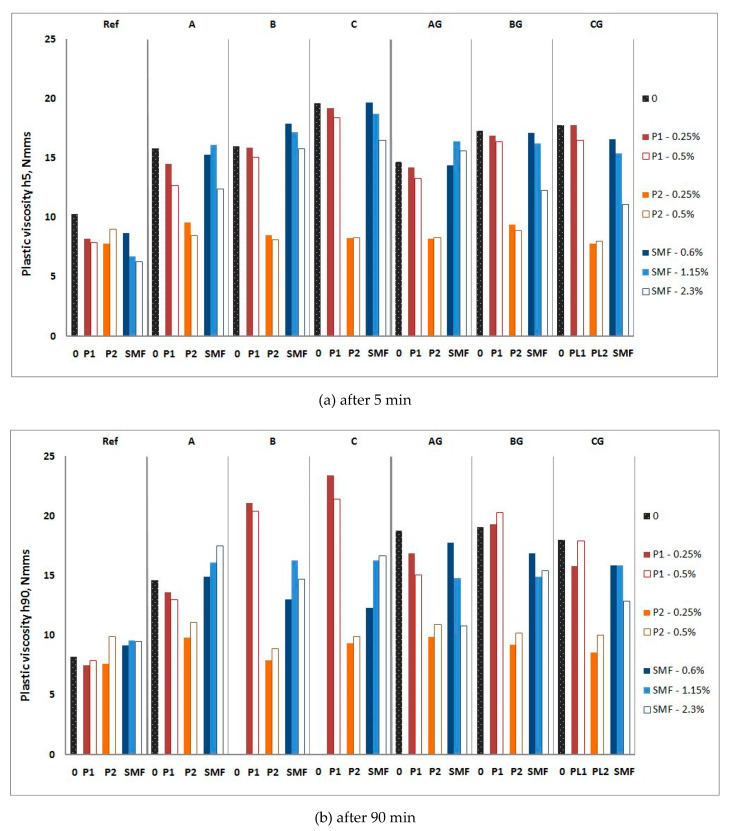
Influence of P1 and P2 and SMF on plastic viscosity h of mortars with raw and ground CFA. (**a**) after 5 min; (**b**) after 90 min.

**Figure 12 materials-13-02245-f012:**
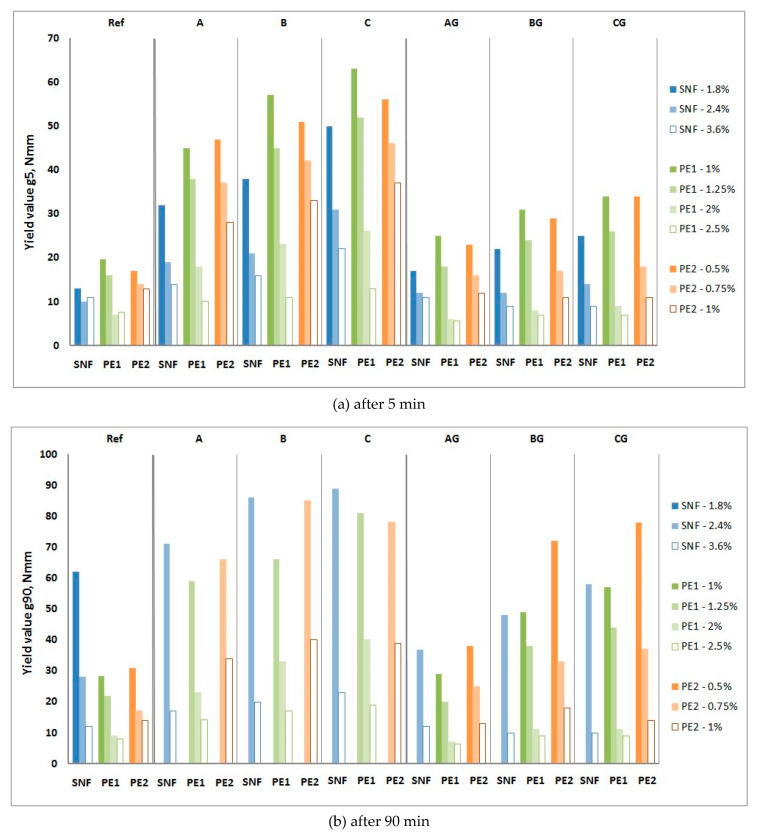
Influence of SNF, PE1 and PE2 on yield value g of mortars with raw and ground CFA. (**a**) after 5 min; (**b**) after 90 min.

**Figure 13 materials-13-02245-f013:**
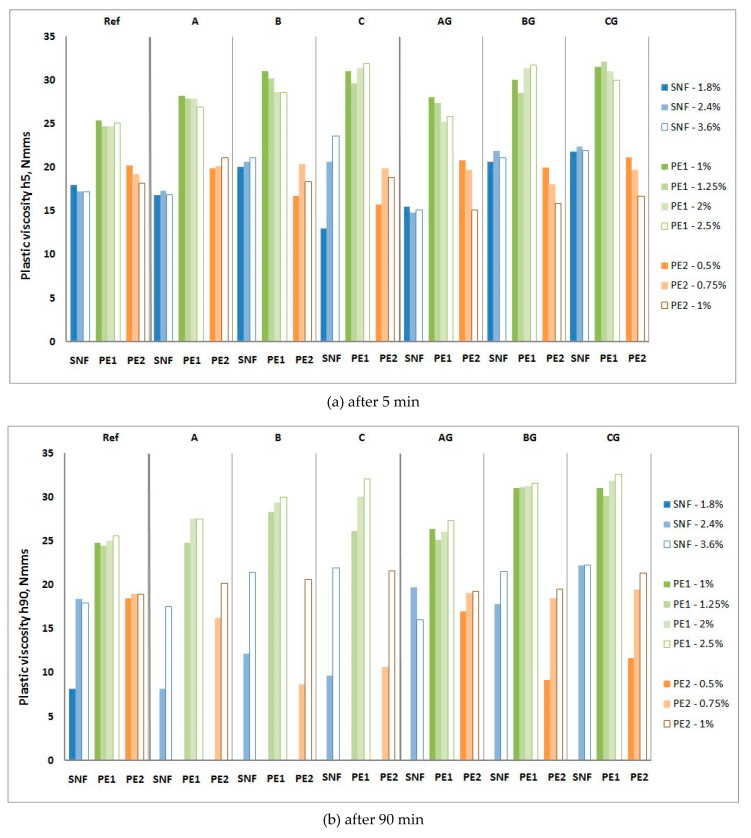
Influence of SNF, PE1 and PE2 on plastic viscosity h of mortars with raw and ground CFA. (**a**) after 5 min; (**b**) after 90 min.

**Figure 14 materials-13-02245-f014:**
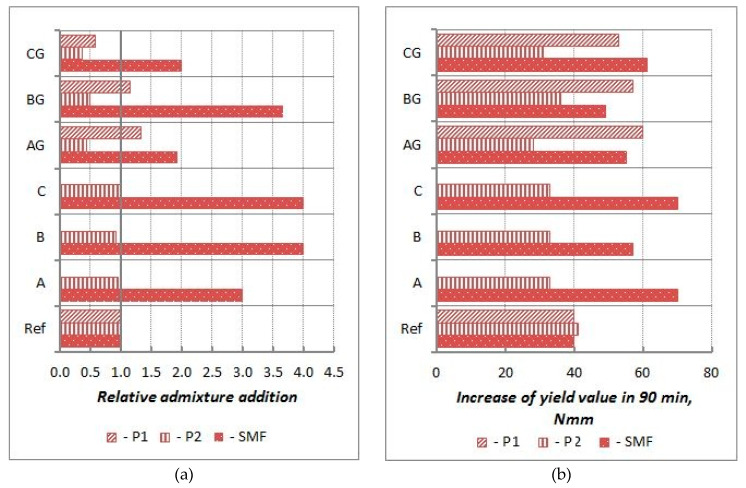
Influence of CFA on effectivness of P1, P2 and SMF (mortars of w/b = 0.55); (**a**) relative admixture content (in relation to reference mortar without CFA) neccessary to be added to obtain mortar with g5 = 20 Nmm; (**b**) increase in yield value g of mortars with initial yield value g equal 20 Nmm in time.

**Figure 15 materials-13-02245-f015:**
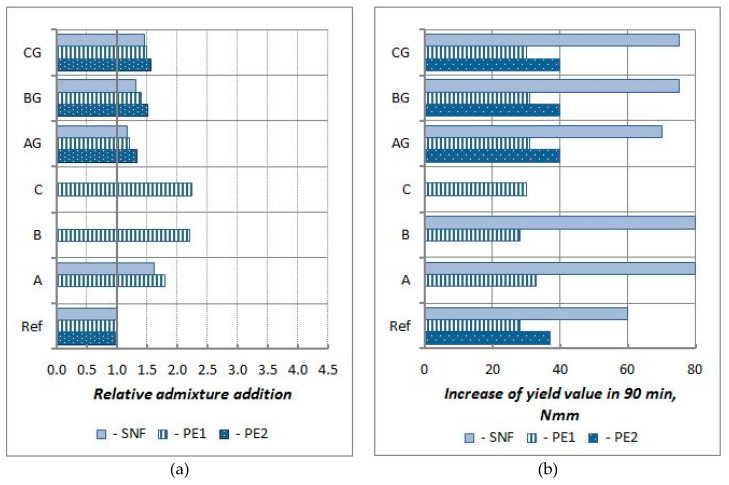
Influence of CFA on effectivness of SNF, PE1 and PE2 (mortars of w/b = 0.45); (**a**) relative admixture content (in relation to reference mortar without CFA) neccessary to be added to obtain mortar with g5 = 20 Nmm; (**b**) increase in yield value g of mortars with initial yield value g equal 20 Nmm in time.

**Figure 16 materials-13-02245-f016:**
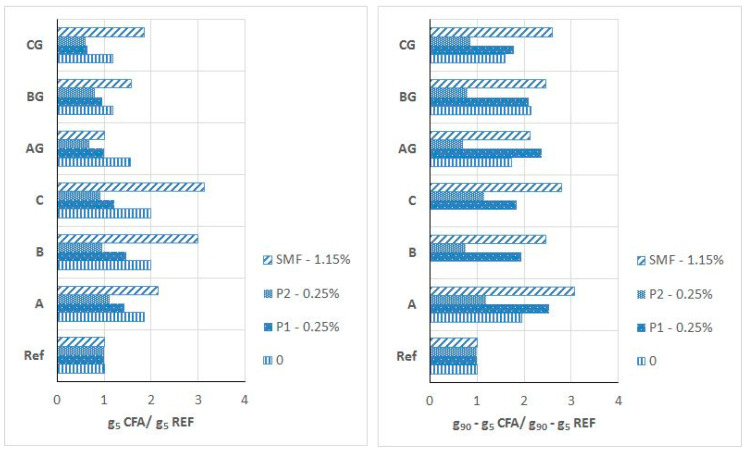
Relative effect of CFA presence on initial yield value g (g5) and yield value g increase in time (g90–g5) of mortars in respect to reference mortars REF without CFA (Mortars without and with P1 or P2 or SMF − ½ of recommended maximum dosage, w/b = 0.55).

**Figure 17 materials-13-02245-f017:**
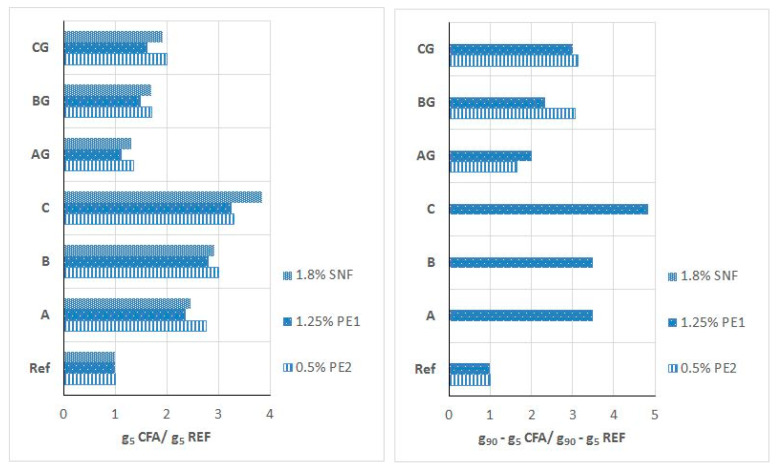
Relative effect of CFA presence on initial yield value g (g5) and yield value g increase in time (g90–g5) of mortars in respect of reference mortars without CFA. (Mortars with SNF or PE1 or PE2 − ½ of recommended maximum dosage, w/b = 0.45).

**Table 1 materials-13-02245-t001:** Types, primary and secondary effects of plasticizers (P) and superplasticizers (SP) [24].

Admixture	Type	Primary Effect	Secondary Effects-Side Effects
P	lignosulfonates and its salts (Ca, Na, Mg, NH4); hydroxy-carboxylate acids and its salts (containing groups (OH), (COOH)).	Influence on rheological properties of cement mixtures enabling: increase in workability (fluidity) of mixture (constant w/b ratio) decrease in w/b ratio at given workability of cement mixture (enabling increase in compressive strength and durability of hardened cement composite) decrease in cement content at given fresh and hardened cement mixture properties	influence on cement setting time influence on air content in mixture influence on heat of cement hydration
SP	salts of sulfonated naphthalene formaldehyde polymers (SNF); salts of sulfonated melamine formaldehyde polymers; (SMF); polycarboxylate acrylic acids polymers and cross-linked polymers (PC and CLPC) polycarboxylate ethers polymers (PE); Rother substances in example modified lignosulfonates.		

**Table 2 materials-13-02245-t002:** Research plan—type of calcareous fly ash (CFA), w/b ratio, admixture dosage and tested properties.

Type and Batches of Calcareous Fly Ash (CFA)	w/b Ratio	Symbol of Admixture		For Testing Rheological Properties [% b.m]	For Testing Air Content [% b.m]	For Testing Heat of Hydration [% b.m]
Raw CFA: A B, C Ground CFA: AG, BG, CG CFA content: 20% as cement mass replacement	0.55	P	P1	0, 0.25, 0.5%	0, 0.25%	x
	0.55		P2	0, 0.25, 0.5%	0, 0.25%	0, 0.25%
	0.55	SP	SMF	0, 0.6, 1.15, 2.3%	0, 1.15%	0, 1.15%
	0.45		SNF	1.8, 2.4, 3.6%	1.8%	0, 1.8%
	0.45		PE1	1.0, 1.25, 2.0, 2.5%	1.25%	0, 1.25%
	0.45		PE2	0.5, 0.75, 1.0%	0.5%	x

**Table 3 materials-13-02245-t003:** Chemical composition of CFA.

CFA	LOI	SiO_2_	Al_2_O_3_	Fe_2_O_3_	CaO	SO_3_	K_2_O	Na_2_O	CaO_w_	Bulk Density [kg/m^3^]	Fineness		Blaine Specific Surface [34] [cm^2^/g]	
											Raw	Ground G	Raw	Ground G
A	2.56	33.47	19.19	5.37	31.18	4.33	0.11	0.31	3.43	1098	36.4	23	2860	3500
B	2.12	40.98	19.00	4.25	25.97	3.94	0.14	0.13	1.07	1028	46.3	20.8	2370	3520
C	2.67	45.17	20.79	4.58	20.6	2.5	0.19	0.23	1.18	960	57.2	16.7	1900	3700

**Table 4 materials-13-02245-t004:** The type, chemical base, density and volume of chemical admixtures. Date obtained from the manufacturer of admixture.

Symbol of Admixture		Chemical Base	Density at 20 °C, [g/cm^3^]	Maximum Recommended Dosage, [% b.m]
P	P1	lignosulfonates	1.00+/−0.01	0.5%
	P2	iminodietanol, bis ethanol, phosphate (V) tri butyl acetate, formaldehyde, methanol, (Z)-octadec-9-enyloamine	1.07+/−0.01	0.5%
SP	PE1	polycarboxylate ether	1.07+/−0.02	2.5%
	PE2	polycarboxylate ether	1.07+/−0.02	1.0%
	SMF	melamine sulfonates	1.20+/−0.03	2.3%
	SNF	naphthalene sulfonate	1.15+/−0.03	3.6%

**Table 5 materials-13-02245-t005:** Properties of cement CEM I 42.5. Data obtained from the cement producer.

SiO_2_ [%]	Al_2_O_3_ [%]	Fe_2_O_3_ [%]	CaO [%]	SO_3_ [%]	Na_2_O_e_ [%]	C_3_S [%]	C_2_S [%]	C_3_A [%]	C_4_AF [%]	Spec. Surf., [34] [cm^2^/g]
20.5	4.89	2.85	63.3	2.76	0.73	65	10	8.1	8.7	3500

**Table 6 materials-13-02245-t006:** Composition of mortars for testing the rheological properties.

Constituent	Amount, [g/batch]
Cement	450/405/360/315
Calcium Fly Ash	-/45/90/135
w/(c + CFA)	0.45/0.55
Water	202.5/247.5
Standard sand	1350

**Table 7 materials-13-02245-t007:** Influence of P and SP on air content in mortars with and without CFA.

CFA	Air Volume [%]						
	without Admixture	0.25% P1	0.25% P2	1.15% SMF	1.8% SNF	1.25% PE1	0.5% PE2
CEM I	5.2	4.6	19.0	2.5	13.5	2.8	9.5
A	2.8	2.4	16.6	3.3	13.1	2.4	13.2
AG	2.5	1.4	15.1	2.3	12.1	2.0	14.4
B	3.5	2.7	16.3	2.6	12.5	3.0	12.3
BG	2.9	2.4	17.5	1.2	11.9	4.0	11.2
C	4.2	2.1	17.2	2.7	12.9	2.0	11.6
CG	2.2	1.7	18.0	1.0	11.0	3.8	10.5

**Table 8 materials-13-02245-t008:** Heat of hydration of cement and CFA paste with P2 and SMF, SNF and PE1 [J/g] during 12 h.

Heat of Hydration, [J/g]							
Sample	10 min	1.5 h	12 h	Sample	10 min	1.5 h	12 h
w/b = 0.55				w/b = 0.45			
CEM I	0.166	3.600	50.315	CEM I	0.177	3.619	51.764
CEM I + ½ max P2	0.011	1.588	20.520	CEM I + ½ max SNF	−0.121	0.904	5.977
CEM I + ½ max SMF	0.122	1.295	19.330	CEM I + ½ max PE1	−0.109	1.313	4.712
A	0.850	7.336	47.157	A	0.753	6.909	47.032
A + ½ max P2	0.885	6.697	32.721	A + ½ max SNF	0.627	5.854	21.384
A + ½ max SMF	0.852	6.277	34.699	A + ½ max PE1	0.650	4.583	14.576
AG	1.433	9.018	49.557	AG	1.230	8.596	69.799
AG + ½ max P2	1.252	7.032	35.273	AG + ½ max SNF	0.991	6.437	21.983
AG + ½ max SMF	1.297	7.139	36.311	AG + ½ max PE1	1.034	6.101	18.514
B	0.876	6.843	44.860	B	0.876	6.843	44.860
B + ½ max P2	0.652	5.212	27.130	B + ½ max SNF	0.853	5.766	19.844
B + ½ max SMF	0.684	5.645	32.767	B + ½ max PE1	0.506	4.462	13.293
BG	1.095	7.994	49.350	BG	1.095	7.994	49.350
BG + ½ max P2	0.715	5.627	30.805	BG + ½ max SNF	0.723	6.021	21.042
BG + ½ max SMF	0.882	5.913	33.511	BG + ½ max PE1	0.765	5.660	17.697
C	0.907	6.905	46.781	C	0.849	6.526	47.159
C + ½ max P2	0.770	5.363	31.955	C + ½ max SNF	0.644	4.716	14.542
C + ½ max SMF	0.578	4.511	27.013	C + ½ max PE1	0.606	3.880	10.974
CG	1.335	7.049	46.999	CG	1.256	6.838	48.463
CG + ½ max P2	1.493	7.253	40.282	CG + ½ max SNF	1.150	6.224	17.704
CG + ½ max SMF	1.105	5.631	32.095	CG + ½ max PE1	0.945	5.244	10.189

**Table 9 materials-13-02245-t009:** Analysis of Variance (ANOVA). One-dimensional significance tests for rheological parameters of mortar with P.

Impact of CFA and P on Rheological Parameters of Mortars	g5 [Nmm]		g90 [Nmm]		h5 [Nmms]		h90 [Nmms]	
	The Value of F	Level of Significance p	The value of F	Level of Significance p	The Value of F	Level of Significance p	The Value of F	Level of Significance p
Raw and ground CFA; type of batches	**55.97577**	**0.000000**	**59.9000**	**0.004932**	**53.69909**	**0.000000**	**25.33230**	**0.000009**
Type of P	1.25945	0.120692	19.5596	1.85693	1.89624	0.190867	1.69624	0.255697
Dosage of P, [% b.m.]	1. 95367	0.100815	13.5963	2.26472	1.69185	0.300257	1.23665	0.30236
Raw and ground CFA; batches and Type of P	**9.68102**	**0.000205**	**56.5476**	**0.00998**	**17.53595**	**0.000009**	**9.16799**	**0.000270**
Raw and ground CFA; batches and Dosage of P, [% b.m.]	1.89637	0.140802	20.0119	1.01307	2.19993	0.093236	1.89624	0.140827
Type of P and Dosage of P, [% b.m.]	2.71871	0.111022	12.5952	3.37841	1.19355	0.336703	1.10185	0.363647

Significant statistical influence is marked in bold italics.

**Table 10 materials-13-02245-t010:** ANOVA. One-dimensional significance tests for rheological parameters of mortar with SP.

Impact of CFA and SP on Rheological Parameters of Mortars	g5 [Nmm]		g90 [Nmm]		h5 [Nmms]		h90 [Nmms]	
	The Value of F	Level of Significance p	The Value of F	Level of Significance p	The Value of F	Level of Significance p	The Value of F	Level of Significance p
Raw and ground CFA; type of batches	**549.774**	**0.0178**	**18.4861**	**0.000027**	**3.937173**	**0.020769**	**4.84901**	**0.010359**
Type of SP	25.263	5.3485	5.7630	0.006398	2.536958	0.096357	2.5693	0.019653
Dosage of SP, [% b.m.]	**1854.023**	**0.000000**	**512.1678**	**0.000000**	**52.2563**	**0.000569**	**23.45659**	**0.000215**
Raw and ground CFA; batches and Type of SP	33.214	8.3383	1.6148	0.188646	3.094208	0.034357	2.33994	0.045490
Raw and ground CFA; batches and Dosage of SP, [% b.m.]	54.996	13.8065	1.7850	0.015218	1.021517	0.498316	2.09643	0.081593
Type of SP and Dosage of SP, [% b.m.]	10.816	25.5603	1.3870	0.025324	1.315541	0.304389	2.76971	0.016045

Significant statistical influence is marked in bold italics.

**Table 11 materials-13-02245-t011:** Influence of raw and ground CFA addition on the technical effectiveness level of P and SP action.

Type of Admixture	In Raw CFA Presence	In Ground CFA Presence
P:	Technical effectiveness of P action	
P1- lignosulfonates (max 0.5%)	−2 *	0 *
P2- iminodietanol, bis ethanol, phosphate (V) tri butyl acetate, formaldehyde, methanol, (Z)-octadec-9-enyloamine (max 0.5%)	2 *	2 *
SP:	Technical effectiveness of SP action	
SMF- melamine sulfonates (max 2.3%)	−2 *	−2 *
SNF- naphthalene sulfonate (max 3.6%)	−2 *	−1 *
PE1- polycarboxylate ether (max 2.5%)	−2 *	0 *
PE2- polycarboxylate ether (max 1.0%)	−2 *	−1 *

Explanation of symbols in the table: *−2- significantly reduced efficiency compared to operation without CFA; −1- slightly reduced efficiency compared to operation without CFA; 0- unchanged efficiency compared to operation without CFA; 1- slightly increased efficiency compared to operation without CFA; 2- significantly increased efficiency compared to operation without CFA.

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
