# Peer review of "The Influence of Calcareous Fly Ash on the Effectiveness of Plasticizers and Superplasticizers"

_materials, 2020, doi:10.3390/ma13102245_

Round 1

Reviewer 1 Report

I think the mauscript has been improved during the different rounds and I think it is almost ready for publishing it. Some minor comments:

  • There is a mistake of the units in bulk density of sand (2650 kg/m3 or 2.65 g/cm3)
  • I still think that the quality of the figures should be improved and personalized. Figures 1, 2 and 3, are not legible, text is very Little for Reading it. I suggest to modify and make them better. I suggest the same for figures 4, 5 and 6. I think those images are directly given as report of testing machine. Make them more readable.
  • In addition, I suggest to join the figures of DRX and TGA in two (one for DRX and other for TGA) in order to reduce the high number of figures in the manuscrpt.

Author Response

Thank you for your sincere review.

I addressed all your comments carefully.

I think the mauscript has been improved during the different rounds and I think it is almost ready for publishing it. Some minor comments:

There is a mistake of the units in bulk density of sand (2650 kg/m3 or 2.65 g/cm3.

- We revised. I still think that the quality of the figures should be improved and personalized. Figures 1, 2 and 3, are not legible, text is very Little for Reading it. I suggest to modify and make them better. I suggest the same for figures 4, 5 and 6. I think those images are directly given as report of testing machine. Make them more readable.

-  All the previous six figures have been corrected by us and presented in a different, more readable form.

In addition, I suggest to join the figures of DRX and TGA in two (one for DRX and other for TGA) in order to reduce the high number of figures in the manuscrpt.

- We corrected the figures.

Reviewer 2 Report

The authors have improved the paper. I have few major comments .

  1. The XRD and DTA curves are need to be re-plotted. The numbers are not clear.
  2. Section 3.3.1 description of rheology with two eq is not necessary. These info are already available in literature. You can delete or reduce the section length with proper citations 
  3. in the results sections, experimental findings are only described. Authors can explain the results with scientific reasons. 
  4. It is mentioned that admixture is strongly absorbed,....But there is no test conducted by the authors to show absorbed %age. 
  5. References are not in same format. Ref. 13. Journal name is not correct.

Author Response

Thank you for your sincere review.

I addressed all your comments carefully.

The authors have improved the paper. I have few major comments.

The XRD and DTA curves are need to be re-plotted. The numbers are not clear.

-  All the previous six figures have been corrected by us and presented in a different, more readable form.

Section 3.3.1 description of rheology with two eq is not necessary. These info are already available in literature. You can delete or reduce the section length with proper citations in the results sections, experimental findings are only described.

– We improved and shortened Section 3.3.1.

Authors can explain the results with scientific reasons.

- We supplemented. Analyzing the effects of P and SP there are no premises to conclude that their mechanism of action is disturbed by CFA. The presence of CFA reduces the amount of water and the amount of active P and SP. These statements, together with relevant literature justifying the scientific reasons, are included in the article.

It is mentioned that admixture is strongly absorbed,....But there is no test conducted by the authors to show absorbed %age.

- We analyzed monographic studies and publications on the effects of SP and P in the presence of ashes. The analysis showed no studies on the effects of admixtures in the presence of calcareous fly ash, but analyzes on silica fly ash are quite numerous. These analyzes indicate the essential importance of the presence of unburned coal and indicate the morphology of the grains, especially their developed surface. This, in our opinion, generally explains the reduced effects of P and SP. Hence our conclusion regarding the absorption of part of SP. This explains the differences in SP effects in raw and ground CFA. SP efficiency is higher at ground CFA, despite the larger Blain specific surface area.

References are not in same format. Ref. 13. Journal name is not correct.

- We corrected.

Reviewer 3 Report

After reading about half of it, I lost my interest.

This paper must be rewritten with the help of a professional proofreader.

Not all figures or tables are necessary. Delete about half of the figures or tables.

There is scientific value in the article but the english is too harsh to follow.

Author Response

Thank you for your sincere review.

I addressed all your comments carefully.

After reading about half of it, I lost my interest.

This paper must be rewritten with the help of a professional proofreader.

- The article was checked by an English proofreader.

Not all figures or tables are necessary. Delete about half of the figures or tables.

- We corrected. Some figures and tables have been removed to make the text more readable.

There is scientific value in the article but the english is too harsh to follow.

- The English language has been checked and corrected by the English proofreader.

Round 2

Reviewer 2 Report

The paper has been now improved as per the comments

Reviewer 3 Report

Thanks for revising the article.

This manuscript is a resubmission of an earlier submission. The following is a list of the peer review reports and author responses from that submission.

Round 1

Reviewer 1 Report

How the Maximum recommended dosage was decided? Why the rheology values are reported only for 5 and 90 mins? Section 4, Please try to explain the results with some scientific concepts. It is no longer a useful information to the readers unless you describe your results. Line 314: why Introduction of CFA increased water demand? Add some papers related to effects of plasticizers and superplasticizers in fly ash concrete and link with your results. Usually addition of fly ash reduces yield stress due to spherical nature of fly ash. So if I replace cement with class F fly ash, I always get a low yield stress material and it saves me from adding superplasticizers to make the concrete more flowable. Why the authors are more focused on effect of plasticizers and superplasticizers in fly ash concrete. This is not clear to me from abstract of this paper. Conclusions is written in discrete manner. So, you can use bullet points to highlight your key findings. A recent paper of fly ash based concrete can be added to improve the literature part. Improving the 3D printability of high volume fly ash mixtures via the use of nano attapulgite clay.

Author Response

Thank you for your valuable comments, we have tried all to apply and parts of the text has been improved.

How the Maximum recommended dosage was decided?

In section 3.1 information about  maximum recommended dosage of PL ad SP have been added. The maximum amount of plasticizers and superplasticizers corresponded to the maximum amount recommended by the producer of admixture. The maximum content of admixture also did not exceed saturation points, which was verified in preliminary studies.

Why the rheology values are reported only for 5 and 90 mins?

The technical effectiveness determines changes in the rheological properties of the fresh concrete in terms of the minimum dosage of admixture needed for its effects to take place, at the intended time needed for transporting and arranging the mix at the installation site, conventionally adopted for 90 min. Therefore, the tests were carried out within the first 90 minutes of mixing the ingredients. Changes were observed during this time.

Section 4, Please try to explain the results with some scientific concepts. It is no longer a useful information to the readers unless you describe your results.

In section 4 some new text fragments have been added.

Line 314: why Introduction of CFA increased water demand?

There are 3 categories of fly ash due to the loss on ignition. The loss on ignition of fly ash consists primarily of particles of unburned carbon in the form of coke breeze. They are mainly shapeless, porous grains with a well-developed specific surface, which makes them very undesirable. The effect of high porosity in the raw CFA is increased water demand with high ash content of loss on ignition. This results in deterioration of the workability of the mix, reduction of the effectiveness of chemical admixtures and reduction of concrete frost resistance.

Add some papers related to effects of plasticizers and superplasticizers in fly ash concrete and link with your results. Usually addition of fly ash reduces yield stress due to spherical nature of fly ash. So if I replace cement with class F fly ash, I always get a low yield stress material and it saves me from adding superplasticizers to make the concrete more flowable.

Papers related to the effects of plasticizers and superplasticizers in concrete were added and combined with results.

Why the authors are more focused on effect of plasticizers and superplasticizers in fly ash concrete. This is not clear to me from abstract of this paper.

The abstract  has been redrafted and changed.

Conclusions is written in discrete manner. So, you can use bullet points to highlight your key findings.

Conclusions have been reworded and corrected.

A recent paper of fly ash based concrete can be added to improve the literature part. Improving the 3D printability of high volume fly ash mixtures via the use of nano attapulgite clay.

Was added.

Reviewer 2 Report

This paper purports to assess the rheological properties of fly ash containing mortars and concretes. Overall, the paper is below-average based on scientific soundness, technical merits, and organization. The Authors have not been able to bring out the novel aspect in the paper. The research justification is not well emphasized. Besides, many important characterization techniques have been evaded which warrant the justification. There are many other issues with the manuscript as well. The paper has been very carelessly prepared and contains several mistakes; for example use of comma instead of decimal points in the numbers, and the  lines 456-476 which should be deleted.

Mortar mix proportions are missing which are very important in ascertaining the and evaluating the conclusions based on the test results. Table 2 does not contain the weight or volume fraction of sand. Details on the properties of sand, e.g. gradation, water absorption, specific gravity, etc. are missing too. These should be provided. What is the justification of Using different water to cement ratios? In order to have a fair comparison, all mixes should have the same w/c ratio. Further, it must be pointed out that in such mixes, instead of water to cement ratio, the use of term “water to binder ratio” is conventional in concrete technology, where binder refers to cement plus supplementary cementing materials. It is advised to modify the water cement ration accordingly. Also, it would be better to provide the quantities in kg/m3 of all the materials of the mix.

XRD of fly ash is very critical as the amorphous or crystalline nature of silica present in it may be responsible for resulting concrete and mortar properties. It is clearly missing here. Please provide.

Brand / mark of plasticizers and super-plasticizers along with the recommended dosages should be mentioned in Table 4. Further, the maximum recommended dosage, [%] mentioned in Table is percentage of what? Is it percent of cement? Or percent of cement + CFA? Please correct the spellings of “recomended” as well.

Particle size of CFA as determined by laser granulometry must be provided. Also, the SEM of CFA particles should be provided. Please note that the particle size and shape/morphology has a significant effect on the theological behavior of resulting

Statstical informationon the result sis missing.

A very critical issue is that major text has been copied from Authors’ own previous work (i) Jacek Gołaszewski, Zbigniew Giergiczny, Tomasz Ponikiewski, Aleksandra Kostrzanowska-Siedlarz, Patrycja Miera. "Effect of Calcareous Fly-ash Processing Methods on Rheological Properties of Mortars", Periodica Polytechnica Civil Engineering, 2018, (ii) Jacek Golaszewski, Aleksandra Kostrzanowska- Siedlarz, Tomasz Ponikiewski, Patrycja Miera. "Influence of Multicomponent and Pozzolanic Cements Containing Calcareous Fly Ash and Other Mineral Admixtures on Properties of Fresh Cement Mixtures", IOP Conference Series: Materials Science and Engineering, 2019, and (iii) Jacek Gołaszewski. "Influence of cement properties on new generation superplasticizers performance", Construction and Building Materials, 2012. This must be rewritten to reduce the similarity index which is 27%, currently.

Author Response

Thank you for your valuable comments, we have tried all to apply and parts of the text has been improved.

Reviewer 3 Report

I think that the paper is interesting but it needs work for its publication. I think that some parts of the manuscript should be rewritten and a statistical analysis is also needed for improving the discussion and to state the conclusions. The manuscript should be line-numbered and page-numbered for the revision round. I cannot count the lines for the comments.

Major and minor comments below.

Abstract: SNF and PE should be previously defined.

Section 1:

Line 3: This sentence needs references.

Line 3: I suggest to change the unit (Mg)

Page 2, first sentence: I suggest to change the connection between the main text and the references [6,7] for making more readable.

Next references [8 and 9] in the paragraph, the authors should be mentioned in the main text.

HCFA: I suggest to provide the differences between CFA and HCFA.

Section 2:

I suggest to include references for the definition of Effectiveness provide in the first line.

Table 1: References are needed in all types and effects.

Section 3. Experimental section.

I suggest re-organizing the section. When table 2 and 3 are mentioned is better to know what mean the symbols A, B, C, G or x.

3.2

Table 4: Include the density of the P and SP.

Reference for  ASTM C618.

“Fluctuations in chemical composition and properties of the ash are significant” statistically? Why?

Method for grinding the HCFA.

20% replacement of cement (in weight?)

Table 7: What type of testing for analyze the cement components? I see the oxides but I also can see the C3s…. DRX? Method used?

Table 8: I suggest to change the table into others with all the different mixtures manufactured to see better the different combinations.

Include the references of the standards used (EN 196-1 PN-EN 1015-7, ….)

3.3.3.

Change 2,5 to 2.5

Results and discussion

There is a sentence “Influence of plasticizers and superplasticizers on rheological properties of HCFA mortars are shown on Fig 1 – 4, and on-air content and heat of hydration in Tables 9 and 10 respectively.” I suggest referencing the figures or tables directly in the discussion. To understand better your discussion, you should include references to figures individually during the discussion in section 4. I think that proper discussion is needed including some reference (not only comparing with other researches) — as you do inline ….. “This is due to the properties of raw CFA and high water demand, which has been confirmed in [18, 31]”.

The same comment for Fig 8 – 11.

I suggest also include a statistical analysis to state the significant differences between mixes (ANOVA or MANOVA tests).

Author Response

Abstract: SNF and PE should be previously defined.

Effectiveness of superplasticizers SNF (the chemical base is naphthalene sulfonate) and PE (the chemical base is polycarboxylate ether) is slightly lower or does not change. Effectiveness of superplasticizer SMF (the chemical base is melamine sulphonates) is significantly lower.

Section 1:

Line 3: This sentence needs references.

Every year, around 5 million tons of CFA goes to landfills and only a small part is recycled [1].

Line 3: I suggest to change the unit (Mg)

Every year, around 5 million tons of CFA goes to landfills and only a small part is recycled [1].

Page 2, first sentence: I suggest to change the connection between the main text and the references [6,7] for making more readable.

In order to effectively use the CFA in cement and concrete technology in [6 and 7] presented a comprehensive and systematic research program on the activity of the CFA, the CFA compatibility with cement and chemical admixtures for concrete.

Next references [8 and 9] in the paragraph, the authors should be mentioned in the main text.

According to Baran, Drożdż, Giergiczny and Garbacik [8, 9] the hydraulic activity is…

HCFA: I suggest to provide the differences between CFA and HCFA.

Calcareous Fly Ash (CFA) - low-calcium;  High Calcareous Fly Ash  (HCFA) – high-calcium

has been explained in the text

Section 2:

I suggest to include references for the definition of Effectiveness provide in the first line.

We added a reference

Table 1: References are needed in all types and effects.

Section 3. Experimental section.

I suggest re-organizing the section. When table 2 and 3 are mentioned is better to know what mean the symbols A, B, C, G or x.

When tables 2 and 3 are mentioned, they have been supplemented with a description explaining where they are chemical composition and properties of batches

…unprocessed fly ash; batches: A, B, C (chemical composition and properties in tab. 5 and 6)

…processed fly ash; batches: AG, BG, CG (chemical composition and properties in tab. 5 and 6)

3.2

Table 4: Include the density of the P and SP.

Admixture

Symbol

Chemical base

Density at 20oC,

 [g/cm3]

Maximum recomended dosage, [%]

Plasticizers

P1

lignosulfonates

1.00+/-0.01

0.5%

P2

iminodietanol, bis ethanol, phosphate (V) tri butyl acetate, formaldehyde, methanol, (Z)-octadec-9-enyloamine

1.07+/-0.01

0.5%

Superplasticizers

PE1

polycarboxylate ether

1.07+/-0.02

2.5%

PE2

polycarboxylate ether

1.07+/-0.02

1.0%

SMF

melamine sulphonates

1.20+/-0.03

2.3%

SNF

naphthalene sulfonate

1.15+/-0.03

3.6%

Reference for  ASTM C618.

“Fluctuations in chemical composition and properties of the ash are significant” statistically? Why?

Method for grinding the HCFA.

20% replacement of cement (in weight?)

The water demand of tested HCFA is high – replacing 20% mass of cement with HCFA causes the water demand to increase from 8 to 12%

Table 7: What type of testing for analyze the cement components? I see the oxides but I also can see the C3s…. DRX? Method used?

Table 8: I suggest to change the table into others with all the different mixtures manufactured to see better the different combinations.

Include the references of the standards used (EN 196-1 PN-EN 1015-7, ….)

3.3.3.

Change 2,5 to 2.5

2.5 done

Results and discussion

There is a sentence “Influence of plasticizers and superplasticizers on rheological properties of HCFA mortars are shown on Fig 1 – 4, and on-air content and heat of hydration in Tables 9 and 10 respectively.” I suggest referencing the figures or tables directly in the discussion. To understand better your discussion, you should include references to figures individually during the discussion in section 4. I think that proper discussion is needed including some reference (not only comparing with other researches) — as you do inline ….. “This is due to the properties of raw CFA and high water demand, which has been confirmed in [18, 31]”.

The same comment for Fig 8 – 11.

I suggest also include a statistical analysis to state the significant differences between mixes (ANOVA or MANOVA tests).

ANOVA statistical analysis showed that the largest statistical effect on rheological parameters, yield value and plastic viscosity, regardless of the time at which the measurement was made, has type of batches. The rheological parameters of mixtures with a plasticizer are also affected by the interaction of type of batches and type of plasticizer. However, the rheological parameters of mixtures with a superplasticizer are affected by the dosage of superplasticizer.

Round 2

Reviewer 1 Report

The manuscript has been reviewed and revised

Author Response

(The authors gave the same response as above.)

Reviewer 2 Report

I have carefully reviewed the revised version. Regretfully, the Authors have not responded to the Reviewers' comments.Point by point acceptance or rebuttal is missing. The paper quality is extremely weak. In fact, the Authors have not improved the manuscript properly , rather tried to circumvent the real issues raised in the first review round.

Therefore, I recommend to decline the manuscript.

Author Response

(The authors gave the same response as above.)

Reviewer 3 Report

The manuscript has improved during the peer review process but it already needs work prior to publication. Below my comments and suggestions. Please check also the similarity between other papers already published.

Some comments: I suggest, again, that the manuscript should be line and page numbered (mandatory) for making easy the comments to the reviewer.

Still, there are some English mistakes and I suggest to check carefully the text and correct them (some examples: page 2 “it make(s) very hard to…”; page 3 “That definition cover(s)…”; page 10 “CFA affect(s) it clearly …”. Some sentences also need commas.

Some minor comments:

Change [6 and 7] into [6, 7]. There are some mistakes with the format of the references. Check them prior to being published.

My previous comment to mention the authors should be rewritten as “According to Baran and Drożdż [8] and Giergiczny and Garbacik [9]…”

The references [6 and 7] in the previous manuscript were both in the book “Road and Bridges” and now …. Are they new? They are numbered equal in the new version of the manuscript and the previous one. Please check all the references.

Table 12 is in the wrong place.

Delete number 40. When Table 1 is presented on page 15.

I see that there are no references in table 1. I think is mandatory. “These findings are not yours”

Include the references of all the standards used (for example for density of plasticizers and superplasticizers, Blaine specific surface).

I still think that you should briefly explain the method of grinding HCFA.

I still also think that the explanation about cement components (oxides and also C3S, C2S, C3A and C4AF) should be briefly explained in order to replicate the research.

Change “Literature” into “References” on page 12

Table 5. Change commas into points for decimals.

Tablica 6 into Table 6.

Author Response

Comments and Suggestions for Authors

The manuscript has improved during the peer review process but it already needs work prior to publication. Below my comments and suggestions. Please check also the similarity between other papers already published.

Some comments: I suggest, again, that the manuscript should be line and page numbered (mandatory) for making easy the comments to the reviewer.

Supplemented.

Still, there are some English mistakes and I suggest to check carefully the text and correct them (some examples: page 2 “it make(s) very hard to…”; page 3 “That definition cover(s)…”; page 10 “CFA affect(s) it clearly …”. Some sentences also need commas.

….

Some minor comments:

Change [6 and 7] into [6, 7]. There are some mistakes with the format of the references. Check them prior to being published.

Corrected. In order to effectively use the CFA in cement and concrete technology in [6, 7] presented

My previous comment to mention the authors should be rewritten as “According to Baran and Drożdż [8] and Giergiczny and Garbacik [9]…”

Corrected. According to Baran and Drożdż [8] and Giergiczny and Garbacik [9] the hydraulic activity is…

The references [6 and 7] in the previous manuscript were both in the book “Road and Bridges” and now …. Are they new? They are numbered equal in the new version of the manuscript and the previous one. Please check all the references.

Thank you. In the previous manuscript they were correct.

Corrected.

Table 12 is in the wrong place.

Corrected.

Delete number 40. When Table 1 is presented on page 15.

Corrected.

I see that there are no references in table 1. I think is mandatory. “These findings are not yours”

Include the references of all the standards used (for example for density of plasticizers and superplasticizers, Blaine specific surface).

Supplemented. Date obtained from the manufacturer of admixture.

…. something about Properties of CFA…

 I still think that you should briefly explain the method of grinding HCFA.

….

I still also think that the explanation about cement components (oxides and also C3S, C2S, C3A and C4AF) should be briefly explained in order to replicate the research.

….

Change “Literature” into “References” on page 12

Corrected.

Table 5. Change commas into points for decimals.

Table 5 and 7 - Corrected.

Tablica 6 into Table 6.

Corrected.

Round 3

Reviewer 2 Report

This paper purports to assess the rheological properties of fly ash containing mortars and concretes. Overall, the paper is below-average based on scientific soundness, technical merits, and organization. The Authors have not been able to bring out the novel aspect in the paper. The research justification is not well emphasized. Besides, many important characterization techniques have been evaded which warrant the justification. There are many other issues with the manuscript as well. The paper has been very carelessly prepared and contains several mistakes; for example use of comma instead of decimal points in the numbers, and the  lines 456-476 which should be deleted.

Mortar mix proportions are missing which are very important in ascertaining the and evaluating the conclusions based on the test results. Table 2 does not contain the weight or volume fraction of sand. Details on the properties of sand, e.g. gradation, water absorption, specific gravity, etc. are missing too. These should be provided. What is the justification of Using different water to cement ratios? In order to have a fair comparison, all mixes should have the same w/c ratio. Further, it must be pointed out that in such mixes, instead of water to cement ratio, the use of term “water to binder ratio” is conventional in concrete technology, where binder refers to cement plus supplementary cementing materials. It is advised to modify the water cement ration accordingly. Also, it would be better to provide the quantities in kg/m3 of all the materials of the mix.

XRD of fly ash is very critical as the amorphous or crystalline nature of silica present in it may be responsible for resulting concrete and mortar properties. It is clearly missing here. Please provide.

Brand / mark of plasticizers and super-plasticizers along with the recommended dosages should be mentioned in Table 4. Further, the maximum recommended dosage, [%] mentioned in Table is percentage of what? Is it percent of cement? Or percent of cement + CFA? Please correct the spellings of “recommended” as well.

Particle size of CFA as determined by laser granulometry must be provided. Also, the SEM of CFA particles should be provided. Please note that the particle size and shape/morphology has a significant effect on the theological behaviour of resulting

Statistical information on the results is missing.

A very critical issue is that major text has been copied from Authors’ own previous work (i) Jacek Gołaszewski, Zbigniew Giergiczny, Tomasz Ponikiewski, Aleksandra Kostrzanowska-Siedlarz, Patrycja Miera. "Effect of Calcareous Fly-ash Processing Methods on Rheological Properties of Mortars", Periodica Polytechnica Civil Engineering, 2018, (ii) Jacek Golaszewski, Aleksandra Kostrzanowska- Siedlarz, Tomasz Ponikiewski, Patrycja Miera. "Influence of Multicomponent and Pozzolanic Cements Containing Calcareous Fly Ash and Other Mineral Admixtures on Properties of Fresh Cement Mixtures", IOP Conference Series: Materials Science and Engineering, 2019, and (iii) Jacek Gołaszewski. "Influence of cement properties on new generation superplasticizers performance", Construction and Building Materials, 2012. This must be rewritten to reduce the similarity index which is 27%, currently.

Author Response

Comments and Suggestions for Authors
This paper purports to assess the rheological properties of fly ash containing mortars and concretes. Overall, the paper is below-average based on scientific soundness, technical merits, and organization. The Authors have not been able to bring out the novel aspect in the paper. The research justification is not well emphasized. Besides, many important characterization techniques have been evaded which warrant the justification. There are many other issues with the manuscript as well. The paper has been very carelessly prepared and contains several mistakes; for example use of comma instead of decimal points in the numbers, and the  lines 456-476 which should be deleted.
Supplemented:

The condition of practical application of CFA in concrete technology is the solution of the workability problem. To obtain required workability of CFA containing concrete it is necessary to use plasticizers (P) or superplasticizers (SP), it can even be said that the possibility of using CFA is conditional on the use of these admixtures [11]. Therefore the issue of effectiveness of the admixtures in the presence of High Calcareous Fly Ash  (HCFA) – high-calcium is particularly important. However, the experimental data on this topic has been limited. In general, to obtain a specific workability of fresh concrete with CFA, it is necessary to use more P or SP than for of the corresponding compositions without CFA [19,20,21,22]. It appears that this is due primarily greater water demand of fresh concrete with CFA - in consequence there is a smaller amount of free water in a mixture. Potentially lower efficiency of P and SP in the presence of HCFA is indicated by the faster loss of workability of mixes with HCFA [11], but such effects does not occur in every case [18]. Generally, this allows to conclude that the present knowledge on the impact of CFA the effectiveness of plasticizers and superplasticizers is not systematic and further studies are needed. There has been no research so far on how different types of P and SP work with CFA with different properties.

The main objectives of this research was to investigate the influence of raw and processed CFA on effectiveness of action of typically used in practice plasticizers and superplasticizers. The basic effect of P and SP on rheological properties was studied, in addition, the secondary effects of such admixture effects as setting time, heat of hydration and air content were also studied. In a broader aspect, the research contributes to popularize possibility of calcareous fly ash use in cement and concrete technology, what greatly benefits the environment protection.

Conclusion: It was found that the presence of ash affects the efficiency of P and SP, while processing by grinding of ash causes that the effect becomes negligible. This is new in both the cognitive and practical aspects.

Mortar mix proportions are missing which are very important in ascertaining the and evaluating the conclusions based on the test results.

Supplemented:

The proportions of mortars mixture were based on standard mortar proportioning according to EN 196-1 but with w/c ratio changed to 0.45 or 0.55. Mortars proportions are shown in Table 8. 

Some of the information about dosage of admixture and research plan are shown in Table 2 and Table 3.

Table 2 does not contain the weight or volume fraction of sand. Details on the properties of sand, e.g. gradation, water absorption, specific gravity, etc. are missing too. These should be provided.

Supplemented:

In order to eliminate the influence of type and grading of sand on rheological properties of mortars,  EN 196-1 CEN normal sand (2 mm max. and bulk density of sand is 2,65 kg/m3) was used. The grading curve of normal sand is presented in Fig. 13.

What is the justification of Using different water to cement ratios? In order to have a fair comparison, all mixes should have the same w/c ratio. Further, it must be pointed out that in such mixes, instead of water to cement ratio, the use of term “water to binder ratio” is conventional in concrete technology, where binder refers to cement plus supplementary cementing materials. It is advised to modify the water cement ration accordingly. Also, it would be better to provide the quantities in kg/m3 of all the materials of the mix.

Corrected: instead of “water to cement ratio” used “water to binder ratio”.

Changed w/b due to the fact that P action is insufficient to liquefy mortars (too high workability) with w/b lower than 0.55, and SP works too strongly and can segregate mortars with w/b greater than 0.45.

Brand / mark of plasticizers and super-plasticizers along with the recommended dosages should be mentioned in Table 4. Further, the maximum recommended dosage, [%] mentioned in Table is percentage of what? Is it percent of cement? Or percent of cement + CFA? Please correct the spellings of “recommended” as well.

 Maximum recommended dosage, [% b.m]

 XRD of fly ash is very critical as the amorphous or crystalline nature of silica present in it may be responsible for resulting concrete and mortar properties. It is clearly missing here. Please provide.

Particle size of CFA as determined by laser granulometry must be provided. Also, the SEM of CFA particles should be provided. Please note that the particle size and shape/morphology has a significant effect on the theological behaviour of resulting.

Supplemented:

The XRD pattern of FCA are presented in Figs.: 1-3. The DTA pattern of FCA are presented in Figs.: 4-6. Cumulative distribution of ash grain size are presented in Figs.: 7-9. This ash contains in the composition, both above 25% reactive silica and above 10% reactive calcium oxide, which shapes its pozzolanohydraulic properties. The results of supplementary tests in terms of phase composition and granulation confirm that the above-mentioned remarks regarding the usefulness of calcareous fly ash as a pozzolan-hydraulic component of cements refer to batches of material with different phase composition (diffractograms and thermograms in Figs.: 1-6) and variable particle size (Figs.: 7-9) within the fluctuations shown during intensive monitoring. Observations of calcareous fly ash using scanning electron microscopy showed the presence of grains with a spherical shape and a smooth surface as well as irregularly shaped porous grains (Figs.: 10-12).

Supplemented:

Research indicates that the presence of large porous, amorphous grains reduces the effectiveness of P and SP, because the admixture is strongly absorbed by the grains, which requires an increased amount of admixture to obtain adequate liquefaction. Removing or grinding these beans removes this problem. Statistical information on the results is missing.

Supplemented statistical information:

The relative standard deviation for parameter of yield value g of mortars maximal and average was respectively 4.8% and 4.4% and for parameter of plastic viscosity h of mortars maximal and average was respectively 5.1% and 4.5%.

Supplemented statistical analysis to state the significant differences between mixes (ANOVA) also:

One-dimensional significance tests for rheological parameters: g5, g90, h5, h90 of mixtures with P and with SP is shown in Tables 13 and 14. Tables 13 and 14 show ANOVA with parameterization with sigma-restrictions and decomposition of effective hypotheses. ANOVA statistical analysis showed that the largest statistical effect on rheological parameters, yield value and plastic viscosity, regardless of the time at which the measurement was made, has type of batches. The rheological parameters of mixtures with a plasticizer are also affected by the interaction of type of batches and type of plasticizer. However, the rheological parameters of mixtures with a superplasticizer are affected by the dosage of superplasticizer.

A very critical issue is that major text has been copied from Authors’ own previous work (i) Jacek Gołaszewski, Zbigniew Giergiczny, Tomasz Ponikiewski, Aleksandra Kostrzanowska-Siedlarz, Patrycja Miera. "Effect of Calcareous Fly-ash Processing Methods on Rheological Properties of Mortars", Periodica Polytechnica Civil Engineering, 2018, (ii) Jacek Golaszewski, Aleksandra Kostrzanowska- Siedlarz, Tomasz Ponikiewski, Patrycja Miera. "Influence of Multicomponent and Pozzolanic Cements Containing Calcareous Fly Ash and Other Mineral Admixtures on Properties of Fresh Cement Mixtures", IOP Conference Series: Materials Science and Engineering, 2019, and (iii) Jacek Gołaszewski. "Influence of cement properties on new generation superplasticizers performance", Construction and Building Materials, 2012. This must be rewritten to reduce the similarity index which is 27%, currently.

Reviewer 3 Report

I do not know why there are some comments or suggestions that have not been taken into account by the authors, there are several mistakes in the English and I gave them an advice but the mistakes still are in the manuscript.

I still think that references are needed in table 1

There are no references for all the methods used

Figures 1 to 12 should be better presented. Figure4 to 6 can be joined in one figure and Figure 7 to 9 can be joined, as well, in only one figure

Figure 2 and 3 are totally incomprehensible, I suggest to include a DRX with the three diffractograms, the peaks identified and the differences between them.

Author Response

Thank you for all comments and devoted time.

Comments and Suggestions for Authors

The manuscript has improved during the peer review process but it already needs work prior to publication. Below my comments and suggestions. Please check also the similarity between other papers already published.

Some comments: I suggest, again, that the manuscript should be line and page numbered (mandatory) for making easy the comments to the reviewer.

Supplemented.

Still, there are some English mistakes and I suggest to check carefully the text and correct them (some examples: page 2 “it make(s) very hard to…”; page 3 “That definition cover(s)…”; page 10 “CFA affect(s) it clearly …”. Some sentences also need commas.

Corrected.

Some minor comments:

Change [6 and 7] into [6, 7]. There are some mistakes with the format of the references. Check them prior to being published.

Corrected. In order to effectively use the CFA in cement and concrete technology in [6, 7] presented

P.S.: Finally, the numbering of the reference has been changed.

My previous comment to mention the authors should be rewritten as “According to Baran and Drożdż [8] and Giergiczny and Garbacik [9]…”

Corrected. According to Baran and Drożdż [8] and Giergiczny and Garbacik [9] the hydraulic activity is…

P.S.: Finally, the numbering of the reference has been changed.

The references [6 and 7] in the previous manuscript were both in the book “Road and Bridges” and now …. Are they new? They are numbered equal in the new version of the manuscript and the previous one. Please check all the references.

Thank you. In the previous manuscript they were correct.

Corrected in manuscript and in reference. Detailed citation of the article was indicated.

P.S.: Finally, the numbering of the reference has been changed.

Table 12 is in the wrong place.

Corrected.

Delete number 40. When Table 1 is presented on page 15.

Corrected.

I see that there are no references in table 1. I think is mandatory. “These findings are not yours”

Of course, thank you. Supplemented.

Include the references of all the standards used (for example for density of plasticizers and superplasticizers, Blaine specific surface).

Supplemented. Date obtained from the manufacturer.

 I still think that you should briefly explain the method of grinding HCFA.

Supplemented. Processed HFCA was created by subjecting unprocessed HFCA to a grinding process in a laboratory ball mill. The residue on the 45μm sieve was taken as the measure of grinding.

I still also think that the explanation about cement components (oxides and also C3S, C2S, C3A and C4AF) should be briefly explained in order to replicate the research.

Data obtained from the cement producer.

Change “Literature” into “References” on page 12

Corrected.

Table 5. Change commas into points for decimals.

Table 5 and 7 - Corrected.

Tablica 6 into Table 6.

Corrected.

There are no references for all the methods used.
Supplemented.

Figures 1 to 12 should be better presented. Figure4 to 6 can be joined in one figure and Figure 7 to 9 can be joined, as well, in only one figure.
Figures 4 - 6 will not be combined, first they will be illegible, and secondly we have no technical capabilities - we have obtained test results in the form of non-editable drawings.
Figure 7 to 9 joined in only one figure.

Figure 2 and 3 are totally incomprehensible, I suggest to include a DRX with the three diffractograms, the peaks identified and the differences between them.

We've improved the figures 2 and 3.

Round 4

Reviewer 2 Report

Due to unsatisfactory response to Reviewers' comments, high plagiarism and lack of novelty, I recommend to decline this manuscript.

Please refer to 

1. Jacek Gołaszewski, Zbigniew Giergiczny, Tomasz Ponikiewski, Aleksandra
Kostrzanowska-Siedlarz, Patrycja Miera. "Effect of Calcareous Fly-ash Processing Methods on Rheological Properties of Mortars", Periodica Polytechnica Civil Engineering, 2018

2. Jacek Gołaszewski, Aleksandra Kostrzanowska- Siedlarz, Tomasz Ponikiewski, Patrycja Miera. "Influence of Cements Containing Calcareous
Fly Ash as a Main Component Properties of Fresh Cement Mixtures", IOP Conference Series: Materials Science and Engineering, 2017

3. Jacek Gołaszewski. "Influence of cement properties on new generation superplasticizers performance", Construction and Building Materials, 2012

4. Jacek Golaszewski, Aleksandra Kostrzanowska- Siedlarz, Tomasz Ponikiewski, Patrycja Miera. "Influence of Multicomponent and Pozzolanic
Cements Containing Calcareous Fly Ash and Other Mineral Admixtures on Properties of Fresh Cement Mixtures", IOP Conference Series: Materials Science and Engineering, 2019

5. Nowoświat A, Gołaszewski J. Influence of the Variability of Calcareous Fly Ash Properties on Rheological Properties of Fresh Mortar with Its Addition. Materials (Basel). 2019 Jun;12(12) . doi:10.3390/ma12121942. PMID: 31212894; PMCID: PMC6632051.